# Energy transport, polar amplification, and ITCZ shifts in the GeoMIP G1 ensemble

Rick D. Russotto[1] and Thomas P. Ackerman[1,2]

[1]Department of Atmospheric Sciences, University of Washington, Seattle, Washington, USA
[2]Joint Institute for the Study of the Atmosphere and Ocean, University of Washington, Seattle, Washington, USA

*Correspondence to:* Rick D. Russotto (russotto@uw.edu)

**Abstract.**

The polar amplification of warming and the ability of the inter-tropical convergence zone (ITCZ) to shift to the north or south are two very important problems in climate science. Examining these behaviors in global climate models (GCMs) running solar geoengineering experiments is helpful not only for predicting the effects of solar geoengineering, but also for understanding how these processes work under increased carbon dioxide ($CO_2$). Both polar amplification and ITCZ shifts are closely related to the meridional transport of moist static energy (MSE) by the atmosphere. This study examines changes in MSE transport in 10 fully coupled GCMs in Experiment G1 of the Geoengineering Model Intercomparison Project, in which the solar constant is reduced to compensate for the radiative forcing from abruptly quadrupled $CO_2$ concentrations. In G1, poleward MSE transport decreases relative to preindustrial conditions in all models, in contrast to the CMIP5 abrupt4xCO2 experiment, in which poleward MSE transport increases. We show that since poleward energy transport decreases rather than increasing, and local feedbacks cannot change the sign of an initial temperature change, the residual polar amplification in the G1 experiment must be due to the net positive forcing in the polar regions and net negative forcing in the tropics, which arises from the different spatial patterns of the simultaneously imposed solar and $CO_2$ forcings. However, the reduction in poleward energy transport likely plays a role in limiting the polar warming in G1. An attribution study with a moist energy balance model shows that cloud feedbacks are the largest source of uncertainty regarding changes in poleward energy transport in mid-latitudes in G1, as well as for changes in cross-equatorial energy transport, which are anticorrelated with ITCZ shifts.

## 1 Introduction

As solar geoengineering, or the artificial cooling of earth by reflecting sunlight, increasingly gains attention as part of a possible strategy to deal with the effects of climate change, two important issues are whether the polar amplification of $CO_2$-induced warming can be fully counteracted, and whether regional precipitation patterns will shift, exacerbating flooding in some areas and drought in others (e.g. Irvine et al., 2010). Polar amplification is the phenomenon in which the poles warm by more than the tropics when atmospheric $CO_2$ concentrations are increased. While the reasons for polar amplification are not yet completely understood, it has already been observed in the instrumental temperature record (*e.g.* Bekryaev et al., 2010) and is a robust behavior in climate models (*e.g.* Holland and Bitz, 2003). Since reflecting sunlight would affect the tropics more strongly

than the poles, it has been hypothesized that it would reduce the meridional temperature gradient (e.g. Keith and Dowlatabadi, 1992), leading to tropical cooling and polar warming under the combined $CO_2$ and geoengineering forcings. We refer to this effect as "residual polar amplification". Early model simulations of solar geoengineering scenarios, involving a simultaneous $CO_2$ increase and solar constant decrease (Govindasamy and Caldeira, 2000; Govindasamy et al., 2003), found that the residual

polar amplification was relatively small, with latitudinal temperature patterns looking much more like the preindustrial climate than the climate under increased $CO_2$ without geoengineering. However, some residual polar amplification still occurs as a robust feature of these types of model experiments (e.g. Kravitz et al., 2013a).

Questions about shifts in regional precipitation under solar geoengineering are motivated by evidence from paleoclimate data and climate model studies that if one hemisphere is preferentially warmed, the intertropical convergence zone (ITCZ)

shifts toward that hemisphere (*e.g.* Broccoli et al., 2006). Model studies have shown that such an effect could occur for changes in many different variables that affect the inter-hemispheric albedo contrast, such as high-latitude ice cover (Chiang and Bitz, 2005), mid-latitude forest extent (Swann et al., 2012; Laguë and Swann, 2016), and tropospheric sulfate aerosol concentration (Hwang et al., 2013). In the context of solar geoengineering, Haywood et al. (2013) demonstrated that preferentially injecting reflective aerosols into one hemisphere shifts the ITCZ towards the other hemisphere, causing drought in some tropical areas.

Even under a hemispherically symmetric solar geoengineering deployment, such as space-based mirrors (represented by a solar constant reduction), ITCZ shifts can still occur (Smyth et al., 2017).

Both polar amplification and ITCZ shifts are closely related to the meridional transport of energy by the atmosphere, which makes atmospheric energy transport an important aspect of the effects of solar geoengineering to study. The sensitivity of the ITCZ, in particular, to energy transport in the atmosphere and ocean in present and future climates has recently been a topic

of great research interest. Studies with slab ocean models with imposed hemispherically asymmetric energy fluxes (e.g. Kang et al., 2008; Yoshimori and Broccoli, 2008) found that an anomalous Hadley circulation transports energy from the warmed hemisphere across the equator, shifting the ITCZ towards the warmed hemisphere (because moisture is transported by the lower branch of the Hadley cell, while energy is transported by the upper branch). However, later studies (e.g. Kay et al., 2016; Hawcroft et al., 2017) found that this effect is substantially weaker in GCM simulations that include a full ocean circulation

coupled to the atmosphere. This is because an anomalous wind-driven ocean circulation develops in response to changes in the atmospheric Hadley circulation, allowing the ocean to do most of the work of transporting the excess energy in one hemisphere across the equator (Green and Marshall, 2017). Still, we can consider the ITCZ position in the framework of the atmosphere and take ocean heat transport and storage into account via surface energy fluxes.

This study builds off the methods of a particular set of studies of atmospheric energy transport in projects in which multiple

global climate models (GCMs) were run for the same radiative forcing scenarios. Hwang and Frierson (2010) showed that poleward transport of atmospheric moist static energy (MSE) increases under increased $CO_2$ concentrations in Coupled Model Intercomparison Project, Phase 3 (CMIP3) models. They used a moist energy balance model (EBM) to attribute the change in MSE transport across 40° latitude to different forcing and feedback terms, and found that cloud feedbacks are responsible for most of the inter-model spread in this quantity. Hwang et al. (2011) found that poleward dry static energy (DSE) transport

in mid-to-high latitudes decreases with warming due to the reduced equator-to-pole temperature gradient (since warming is

amplified at the poles), but moisture transport increases due to the overall warming combined with the nonlinearity of the Clausius-Clapeyron equation and the increase in moisture transport is enough to lead to an increase in MSE transport overall. Frierson and Hwang (2012) found that shifts in the ITCZ with warming in the CMIP3 slab ocean ensemble are anticorrelated with changes in atmospheric energy transport across the equator, with cloud feedbacks again being the largest source of uncertainty.

The Geoengineering Model Intercomparison Project (GeoMIP) provides an opportunity to use similar methods to investigate how atmospheric energy transport may change under solar geoengineering conditions, which can help us understand the reasons for residual polar amplification and tropical precipitation shifts. We analyze the results of GeoMIP experiment G1 (Kravitz et al., 2011), in which the solar constant is reduced at the same time as the $CO_2$ concentration is quadrupled in order to maintain top of atmosphere (TOA) energy balance and therefore keep the global mean temperature approximately at preindustrial levels. For comparison, we also examine two CMIP5 model runs: piControl, a preindustrial run, and abrupt4xCO2, in which $CO_2$ is quadrupled but the solar constant remains the same.

GeoMIP has yielded insights into how temperature and precipitation patterns may change under solar geoengineering. Kravitz et al. (2013a) found that residual polar amplification occurs in all models running the G1 experiment. Global mean precipitation decreases in G1 because the reduction in solar radiative flux at the surface is primarily balanced by a reduction in evaporation (Kravitz et al., 2013b). The precipitation reduction is evident over monsoonal land regions, as well as in the global mean, and extreme precipitation events are reduced in G1, in contrast to an increase in abrupt4xCO2 (Tilmes et al., 2013).

Reducing the solar constant is an approximation of more realistic proposals for solar geoengineering, such as injecting aerosols into the stratosphere. One example of how the physics differs between these scenarios involves the magnitude of the reduction in global mean precipitation. Solar constant reduction experiments underestimate the precipitation reduction compared to model runs that increase the concentration of sulfate aerosols in the stratosphere (Niemeier et al., 2013; Ferraro and Griffiths, 2016). This is because sulfate aerosols absorb longwave radiation, which reduces the net atmospheric radiative cooling rate and therefore allows for less precipitation, since the latent heat release from precipitation formation must be balanced by net radiative cooling. (See, *e.g.*, Pendergrass and Hartmann (2014) for a discussion of the atmospheric energy constraints on global mean precipitation). Even with this caveat, the G1 experiment is useful because of its simplicity and because it eliminates confounding effects from global mean temperature changes.

We use the G1 experiment to investigate changes in meridional energy transport under solar geoengineering, the factors responsible for these changes, and the associated effects on tropical precipitation and polar amplification. Section 2 describes the energy and latent heat transport changes that occur in the G1 and abrupt4xCO2 experiments. Section 3 attributes these energy transport changes to different forcing and feedback terms, using the moist energy balance model (EBM) of Hwang and Frierson (2010). Conclusions are provided in Section 4.

## 2 Energy and moisture transport changes

Using the TOA and surface energy and moisture fluxes from the GeoMIP GCM experiments, we calculate changes in the meridional transport of MSE by the atmosphere. By subtracting the MSE transport in piControl from that in G1, we can understand how the combined $CO_2$ increase and solar constant decrease affect MSE transport, or, in other words, how well solar reductions can restore preindustrial patterns of atmospheric energy transport. For comparison, we also examine how the $CO_2$ increase alone affects MSE transport by subtracting the MSE transport in piControl from that in abrupt4xCO2. Following Kravitz et al. (2013a) and other papers analyzing GeoMIP results, we use the average of years 11-50 of the G1 and abrupt4xCO2 experiments in order to exclude the rapid changes that occur during the first few years. For piControl, we average over the first 40 years of model output provided by the modeling groups. (The period analyzed does not include any model spin-up period.) All averages are multi-annual means including all months.

We calculate the energy flux into the atmosphere from the GCM output as the sum of terms on the right-hand side of the following energy budget equation:

$$\nabla \cdot F_M = R^{\downarrow}_{\mathrm{net,TOA}} + R^{\uparrow}_{\mathrm{net,surface}} + \mathrm{SH} + \mathrm{LH} \tag{1}$$

where $F_M$ is the vertically integrated horizontal moist static energy flux, $R^{\downarrow}_{\mathrm{net,TOA}}$ is the net downward radiative flux at the top of the atmosphere, $R^{\uparrow}_{\mathrm{net,surface}}$ is the net upward radiative flux at the surface, SH is the net upward surface sensible heat flux, and LH is the net upward surface latent heat flux. Since the net energy flux into the atmospheric column must be balanced by energy transport out of the column, we can calculate the northward atmospheric moist static energy transport as a function of latitude by integrating $\nabla \cdot F_M$, first zonally and then cumulatively northward from the south pole. It is also useful to decompose the MSE transport into latent energy (moisture) transport and dry static energy (DSE) transport, in order to identify how moisture changes affect the total energy transport changes, following the methodology of Hwang et al. (2011). We calculate the latent energy transport from the latent heat flux and precipitation GCM output by integrating the following equation zonally and meridionally:

$$\nabla \cdot F_L = \mathrm{LH} - L_v P \tag{2}$$

where $F_L$ is the vertically integrated horizontal latent energy flux, $L_v$ is the latent heat of vaporization of water and $P$ is precipitation. We calculate the DSE transport by subtracting the latent energy transport from the MSE transport.

Table 1 lists the GeoMIP models included in this analysis. The models in this study all had solar constant reductions between 3.2% and 5.0% and global mean temperatures in G1 within 0.3 K of those in piControl. Three of the original 13 GeoMIP models are excluded: BNU-ESM, because it did not adequately restore the global mean temperature in the G1 realizations that were available when our analysis was done; EC-Earth, because the precipitation output file was corrupted; and HadCM3, because many of the output fields required for this study are no longer available. Methodological details of these calculations are described in Appendix A.

**Table 1.** Models included in this study, with references, institutions, solar constant reduction in the G1 experiment ($\Delta S_0$), and global mean temperature change in G1 - piControl ($\Delta T$).

| Model | Reference | Institution | $\Delta S_0$ | $\Delta T$ (K) |
|---|---|---|---|---|
| CanESM-2 | Arora et al. (2011) | Canadian Centre for Climate Modeling and Analysis | 4.0% | -0.013 |
| CCSM4 | Gent et al. (2011) | National Center for Atmospheric Research | 4.1% | 0.233 |
| CESM-CAM5.1-FV | Hurrell et al. (2013) | National Center for Atmospheric Research | 4.7% | -0.157 |
| CSIRO-Mk3L-1-2* | Phipps et al. (2011) | Commonwealth Scientific and Industrial Research Organization/ Bureau of Meteorology | 3.2% | 0.034 |
| GISS-E2-R* | Schmidt et al. (2014) | NASA Goddard Institute for Space Studies | 4.5% | -0.292 |
| HadGEM2-ES | Collins et al. (2011) | Met Office Hadley Centre | 3.9% | 0.241 |
| IPSL-CM5A-LR | Dufresne et al. (2013) | Institut Pierre Simon Laplace | 3.5% | 0.109 |
| MIROC-ESM | Watanabe et al. (2011) | Atmosphere and Ocean Research Institute (The University of Tokyo), National Institute for Environmental Studies, and Japan Agency for Marine-Earth Science and Technology | 5.0% | -0.065 |
| MPI-ESM-LR | Giorgetta et al. (2013) | Max Planck Institute for Meteorology | 4.7% | -0.011 |
| NorESM1 | Bentsen et al. (2013) | Bjerknes Centre for Climate Research, Norwegian Meteorological Institute | 4.0% | -0.044 |

* These models are excluded from the second part of this study (the EBM analysis) because the necessary output fields were not archived.

## 2.1 Poleward transport in mid-latitudes

Changes in the zonal mean northward MSE, latent energy and dry static energy (DSE) transport for G1 minus piControl are shown in Figure 1. Poleward MSE transport in mid-latitudes is reduced under G1 in all 10 models and, when decomposed into the latent and DSE components, both of these terms are reduced as well. Figure 2 shows the same calculations for abrupt4xCO2 minus piControl. In this case, poleward DSE transport decreases but moisture transport increases by more than enough to compensate, leading to an increase in total MSE transport. This corroborates the result seen by Held and Soden (2006) and Hwang et al. (2011) in CMIP3 global warming scenarios.

To understand why the energy transport changes are different for global warming and geoengineering conditions, we look at zonal mean changes in temperature and saturation vapor pressure. Figures 3a,b show the zonal mean temperature change in G1 and abrupt4xCO2 relative to preindustrial. (These are also plotted in Figure 1 of Kravitz et al. (2013a), but our plots include only the models analyzed here and, for G1, we use a smaller $y$-axis range to show more detail.) In G1, the tropics are cooled while the poles are warmed (Figure 3a). There is warming everywhere in abrupt4xCO2 but more so in the polar regions, especially the Arctic (Figure 3b). In both cases, this pattern of temperature change results in a weakening of the equator-to-pole temperature gradient and reduces the poleward transport of dry static energy (Figures 1c,f and 2c,f), similar to the result found by Hwang et al. (2011).

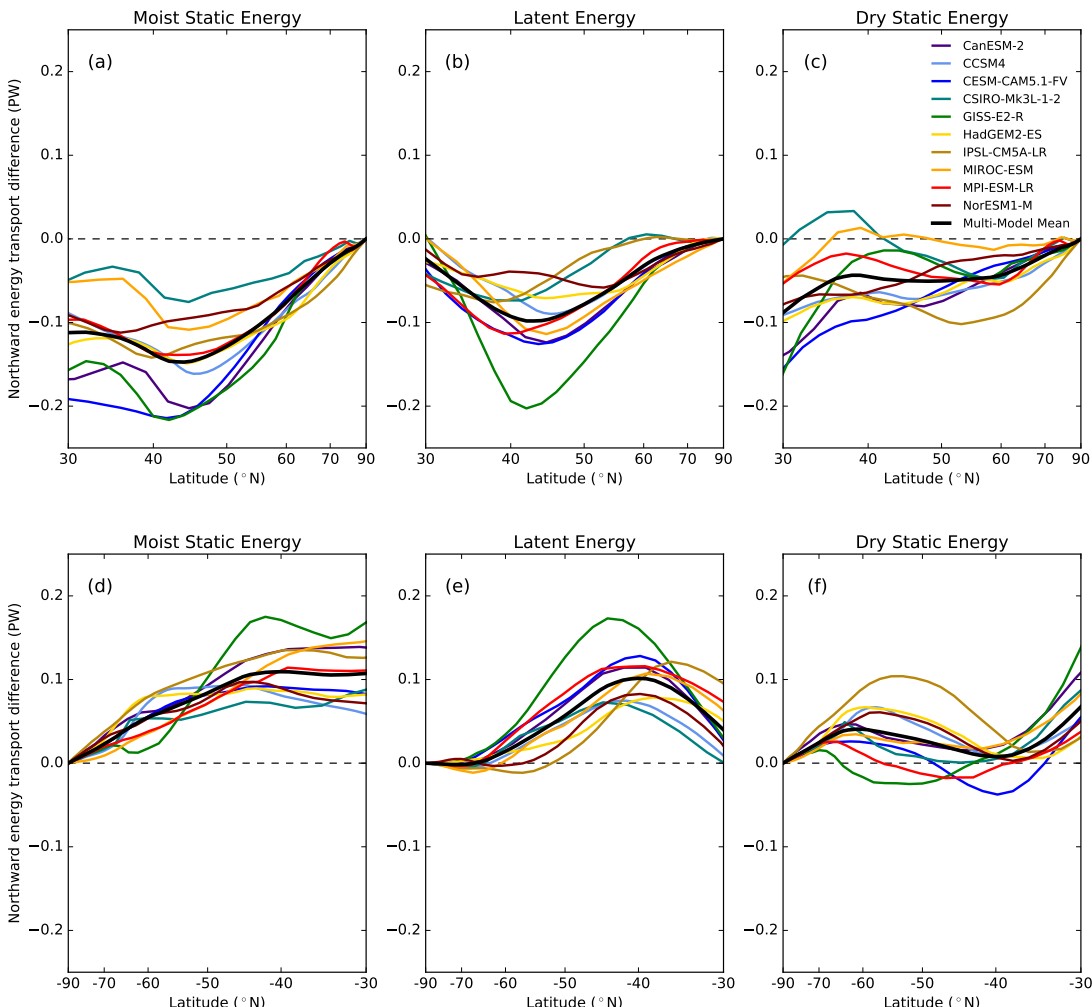

**Figure 1.** Northward energy transport (PW) for G1 minus piControl, poleward of 30° N (a-c) and 30° S (d-f), for total moist static energy transport (a, d), latent energy transport (b, e), and dry static energy transport (c, f), in the GeoMIP models and the multi-model mean.

The mechanism for the difference in moisture transport is apparent from changes in saturation vapor pressure, $e_s$, which we calculate using an approximate form of the Clausius-Clapeyron equation (*e.g.* Hartmann, 2016, eq. 1.11):

$$e_s = (6.11\text{hPa}) \exp \left\{ \frac{L_v}{R_v} \left( \frac{1}{273} - \frac{1}{T} \right) \right\} \tag{3}$$

where $L_v$ is the latent heat of vaporization of water, $R_v$ is the gas constant for water vapor, and $T$ is the temperature in

5   K. In abrupt4xCO2 (Figure 3d), $e_s$ increases more in the tropics than it does near the poles because $e_s$ is approximately

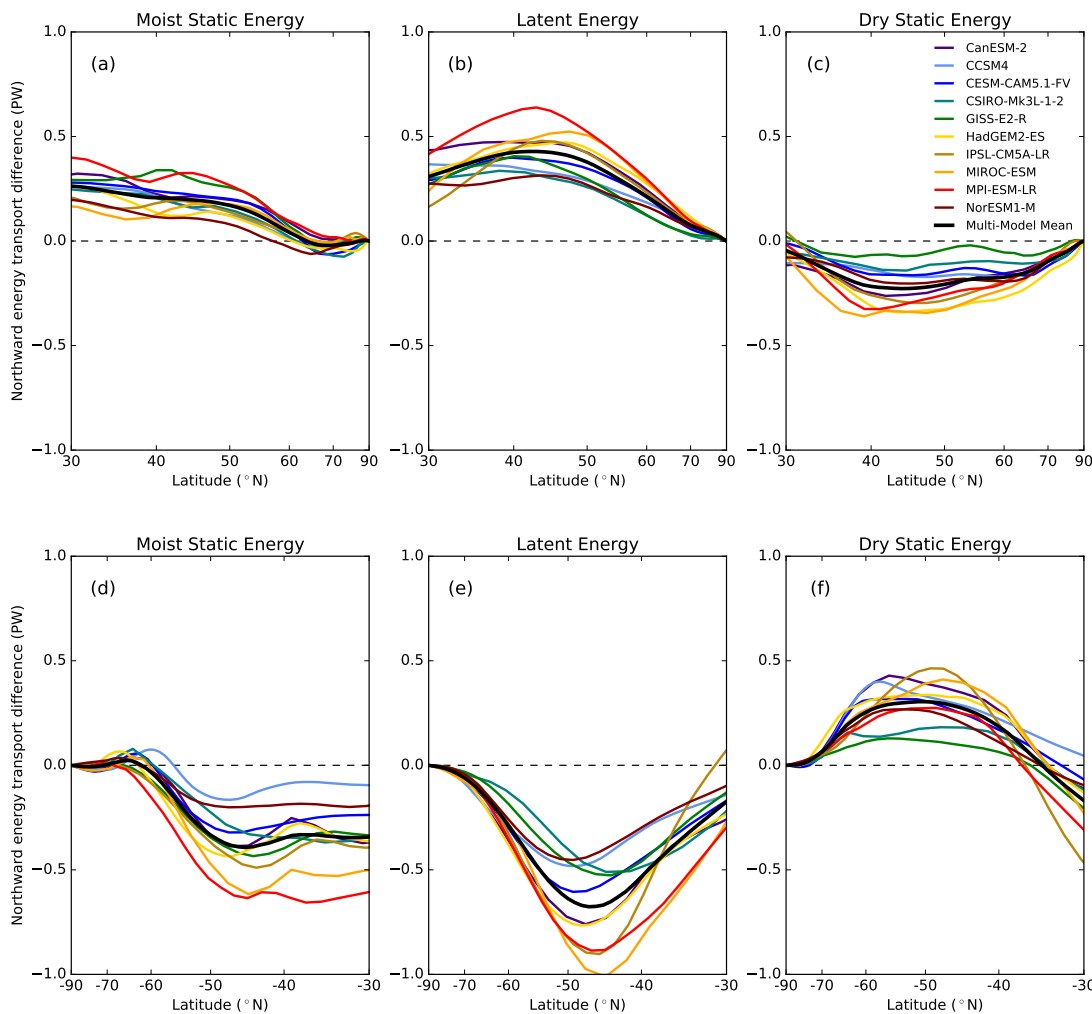

**Figure 2.** As in Figure 1 but for abrupt4xCO2 minus piControl.

exponential with respect to temperature. A slight warming in the tropics leads to a larger increase in $e_s$ because the tropics were initially warmer than the poles. In G1, however (Figure 3c), the tropics cool and the poles warm, so saturation vapor pressure, like temperature, decreases in the tropics and increases (to a lesser extent) at high latitudes relative to piControl. Assuming the moisture content in the atmosphere scales with Clausius-Clapeyron, and the meridional winds are roughly the same, poleward moisture transport increases in abrupt4xCO2 and decreases in G1 because the equator-to-pole moisture gradient has strengthened in abrupt4xCO2 and weakened in G1.

A decrease in poleward energy transport has been previously reported in a single-model study running the G1 setup (Schaller et al., 2014). In addition to a G1 experiment, that study also included runs which included only the $CO_2$ increase or solar

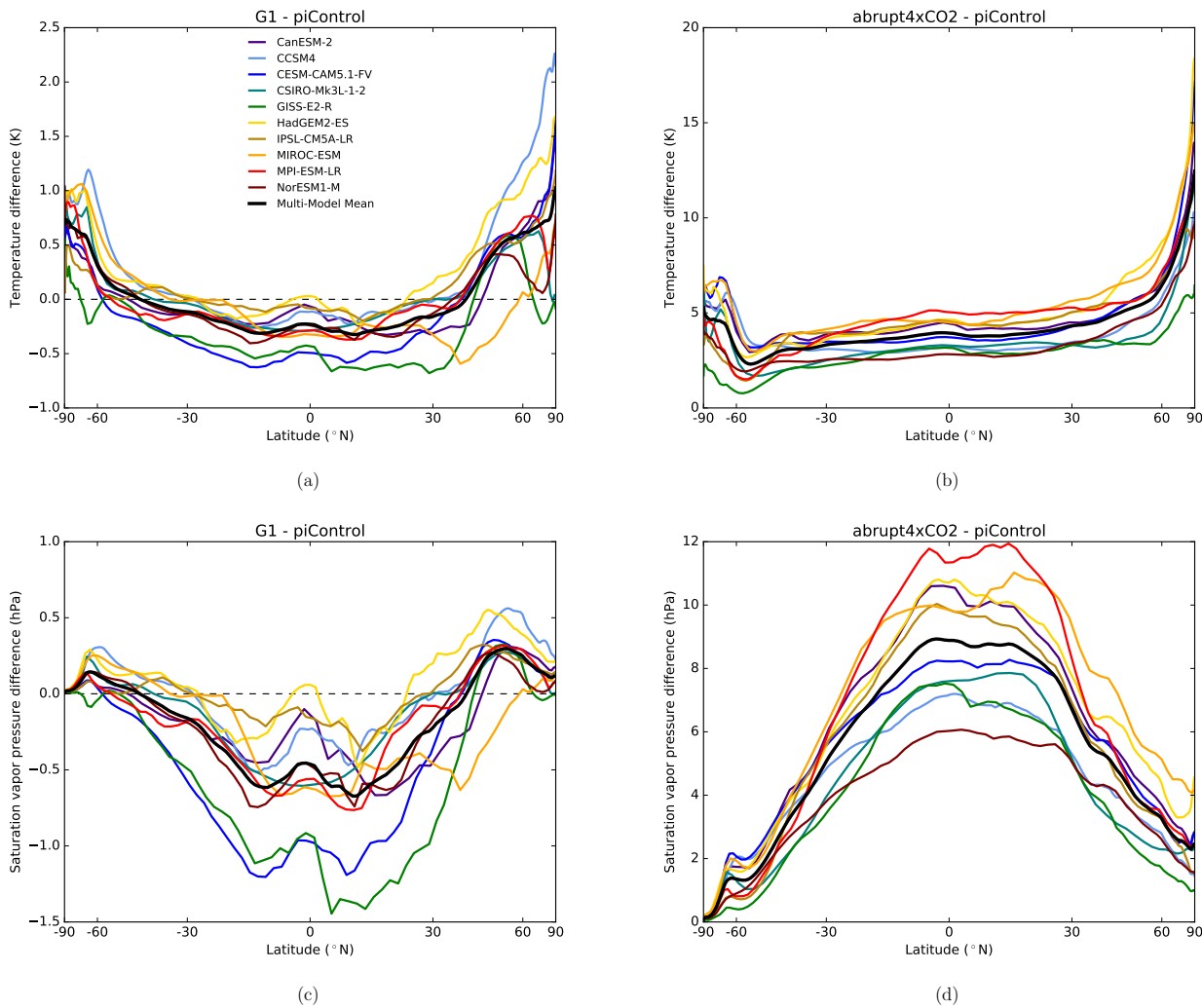

**Figure 3.** Zonal mean surface air temperature changes relative to piControl for G1 (a) and abrupt4xCO2 (b), and analogous zonal mean surface saturation vapor pressure changes (c,d), in GeoMIP models and multi-model mean.

constant decrease. The decrease in poleward energy transport in their G1 run is not equal to the sum of the increase in the "CO2+" run and the decrease in the "solar-" run in either hemisphere (see their Table 3). Also, the changes in poleward energy transport are not symmetrical in their solar increase and solar decrease run. This indicates that the response of meridional MSE transport to various climate forcings is nonlinear, and we cannot simply add and subtract the responses to individual climate forcings to predict the responses to combined forcings. The nonlinearity of the Clausius-Clapeyron equation is a likely source of these nonlinear responses.

In G1, poleward MSE transport decreases, but the poles are still warmed relative to the tropics. This implies that the residual polar amplification in G1 must be due to the differing spatial patterns of the opposing solar and $CO_2$ forcings, with the solar forcing being greater in absolute magnitude in the tropics, where there is more sunlight to reduce. Local radiative feedbacks such as the ice-albedo feedback cannot be responsible for the residual polar amplification because these can only amplify or dampen a temperature change, but cannot reverse its sign. Another possible explanation would be an increase in ocean heat transport; we have not calculated this explicitly, but Hong et al. (2017) found that the Atlantic Meridional Overturning Circulation, which transports heat to the Arctic, slightly weakens in G1. While they only looked at heat transport in the Atlantic, there is a net decrease in energy flux into the ocean in the tropics (see their Figure 4b), so there is no reason to expect an increase in poleward ocean heat transport. In addition, the poleward energy transport by the oceans is much less than that by the atmosphere at high latitudes (e.g. Hawcroft et al., 2017, Figure 6), so small changes to it would not be expected to significantly affect the polar warming. This leaves the differing spatial patterns of the forcings as the only possible explanation for the polar warming and tropical cooling in G1.

The decrease in poleward MSE transport likely diminishes the polar warming in G1, relative to what would happen if forcings and feedbacks were allowed to operate locally but energy transport was fixed at piControl levels. It would be a useful avenue for future research to quantify the effect of reduced energy transport, as well as local feedbacks, on the polar warming in G1, similar to the study of Arctic amplification under global warming by Pithan and Mauritsen (2014). The reduction in poleward moisture transport may help explain the reduction in mid-latitude precipitation seen in Tilmes et al. (2013); quantifying this effect would also be a useful research direction.

## 2.2 Cross-equatorial energy transport and ITCZ shifts

Figure 4a shows the relationship between shifts in the ITCZ in G1 relative to piControl and changes in the transport of moist static energy across the equator. We define the position of the ITCZ as the latitude where half of the zonally integrated rainfall between 15°S and 15°N lies to the south and half lies to the north, following Hwang and Frierson (2010). In the multi-model mean, the ITCZ shifts northward by 0.14 degrees, but there is significant inter-model spread, ranging from -0.33 to 0.89 degrees. A multi-model mean northward ITCZ shift is consistent with the result of Viale and Merlis (2017) that the ITCZ shifts northward by a greater amount for a $CO_2$ increase than for solar constant increase in slab ocean aquaplanet GCM runs with a prescribed northward ocean heat transport that keeps the ITCZ in the northern hemisphere. However, caution must be taken in assuming that the results from $CO_2$ and solar forcing runs can be added linearly, for reasons discussed above. The ITCZ shifts in G1 are moderately anticorrelated with the change in cross-equatorial energy transport (correlation coefficient $r = -0.77$). Anticorrelation between these quantities is consistent with previous work (e.g. Frierson and Hwang, 2012; Hwang et al., 2013), and is expected because the Hadley cell transports energy primarily in its upper branch but moisture primarily in its lower branch.

For ITCZ shifts in abrupt4xCO2, however, there is no correlation between shifts in the precipitation-median ITCZ and cross-equatorial energy transport ($r = 0.07$; not shown). There are several possible reasons for this, and for the fact that some models have close to zero change in cross-equatorial energy flux but nonzero ITCZ shifts in Figure 4a. First, the ITCZ is more closely

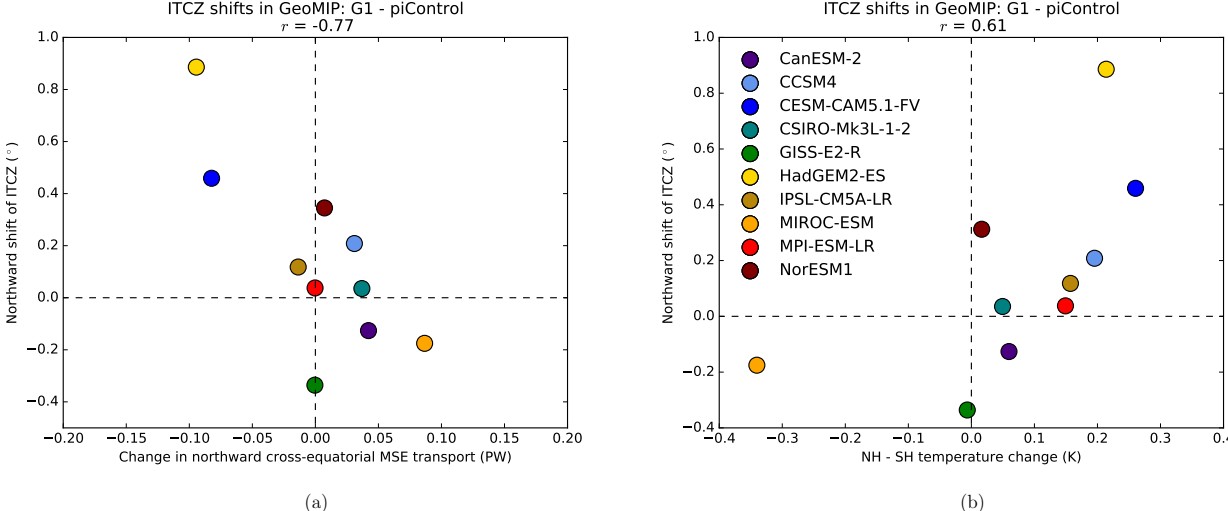

**Figure 4.** Shift in the ITCZ in GeoMIP models for G1 minus piControl, plotted against change in northward MSE transport across the equator (a) and northern hemisphere mean surface temperature change minus southern hemisphere mean temperature change (b). The quantity $r$ is the correlation coefficient.

connected to the "energy flux equator", or the latitude at which the meridional transport of energy by the atmosphere is zero, than it is to the cross-equatorial energy flux. Bischoff and Schneider (2014) developed a theory for the relationship between cross-equatorial energy transport and the energy flux equator, assuming the latter was correlated with the ITCZ position, and argued that while the energy flux equator is proportional to the cross-equatorial energy flux, the constant of proportionality is

governed by the net energy input into the tropical atmosphere, which can allow the energy flux equator to move while cross-equatorial energy transport does not change. We might expect this effect to occur when the atmosphere is suddenly thrown far out of energy balance, as happens when $CO_2$ is abruptly quadrupled. Furthermore, the ITCZ, when defined as the precipitation median or maximum, does not necessarily always follow the energy flux equator, because the the gross moist stability, or the efficiency with which the Hadley circulation exports energy, can change (Seo et al., 2017).

Figure 4b shows the ITCZ shifts in G1 plotted against the warming of the northern hemisphere relative to the southern hemisphere in the same experiment. This is similar to Figure 7 of Smyth et al. (2017), but for the specific set of models included in this study. Here, there is a positive correlation ($r = 0.61$), slightly weaker than that for cross-equatorial MSE transport. While Haywood et al. (2013) found ITCZ shifts away from the cooled hemisphere in the extreme case of aerosol injections in only one hemisphere, the ITCZ shifts in Figure 4b imply that solar reductions applied equally in both hemispheres could still cause

regional shifts in precipitation based on factors like the base state albedo and local radiative feedbacks that might warm one hemisphere relative to the other. This, along with reductions in the seasonal ITCZ migration due to preferential cooling of the summer hemisphere, is discussed elsewhere (Smyth et al., 2017). Our focus here is on the reasons for the inter-model spread in

the ITCZ shifts. These are more easily diagnosed using energy balance model attribution experiments aimed at understanding the causes for the changes in cross-equatorial energy flux.

## 3 Attribution of changes using a moist EBM

In order to investigate the causes of robust changes in meridional energy transport in the G1 experiment and the largest sources of inter-model spread, we ran attribution experiments in which we perturbed different forcing and feedback terms one at a time. These experiments involved the moist energy balance model (EBM) first used by Hwang and Frierson (2010). We used the GCM output to calculate the magnitude of various forcings and feedbacks, including the greenhouse and solar forcings and cloud, surface albedo and non-cloud atmosphere feedbacks, and we used the EBM to understand how atmospheric energy transport would respond to each forcing or feedback in isolation. The advantage of using a moist EBM over directly integrating the energy fluxes associated with each forcing or feedback is that it allows for a coupled response between the energy transport, local temperature, and longwave radiative cooling.

The EBM takes as input the zonal mean surface and TOA energy fluxes and the LW cloud radiative effect from each GCM, and calculates the outgoing longwave radiation (OLR) as a function of surface temperature based on a linear fit of the clear-sky OLR and surface temperature output from each GCM. The net atmospheric energy flux input term is perturbed to account for the influence of various individual forcings and feedbacks, while the intercept of the clear-sky OLR-surface temperature relationship is re-fit in the perturbation climates (G1 and abrupt4xCO2) to account for the enhanced greenhouse effect. Net vertical energy flux convergences at each latitude are balanced by meridional diffusion of MSE. We obtained a meridional energy transport estimate from the EBM by meridionally integrating this diffusion term. Several limitations of the EBM experiments must be noted. The approach of prescribing energy perturbations associated with feedbacks that are static in time does not take into account the interactions of different feedbacks with each other (analyzed by Feldl et al. (2017)) or changes in the feedbacks that arise from the changing energy transport (Merlis, 2014; Rose et al., 2014; Rose and Rayborn, 2016). Also, changing the intercept of the OLR-temperature fit does not account for the nonuniformity of the $CO_2$ radiative forcing with latitude (Huang and Zhang, 2014). Appendix B describes the methods for these experiments in more detail.

### 3.1 Comparison of EBM and GCM-derived energy transport

In order to be helpful in understanding the causes of the GCM behaviors, the EBM needs to predict changes in GCM-derived energy fluxes reasonably well when all forcing and feedback terms are considered. Figure 5 shows the meridional MSE transport across specific latitudes calculated by the EBM versus the same quantities diagnosed directly from the output of each GCM, for cross-equatorial transport in G1 minus piControl (Figure 5a), for poleward transport across 40°N/S in G1 minus piControl (Figure 5b), and for the same comparison in abrupt4xCO2 minus piControl (Figure 5c). There are generally strong correlations in each of these cases. Cross-equatorial energy transport changes in abrupt4xCO2 minus piControl are not examined because these did not correlate well with ITCZ shifts. Note that, for G1 minus piControl, the cross-equatorial energy transport appears to change more easily in the GCMs than in the EBM, while the poleward energy transport across 40° N/S

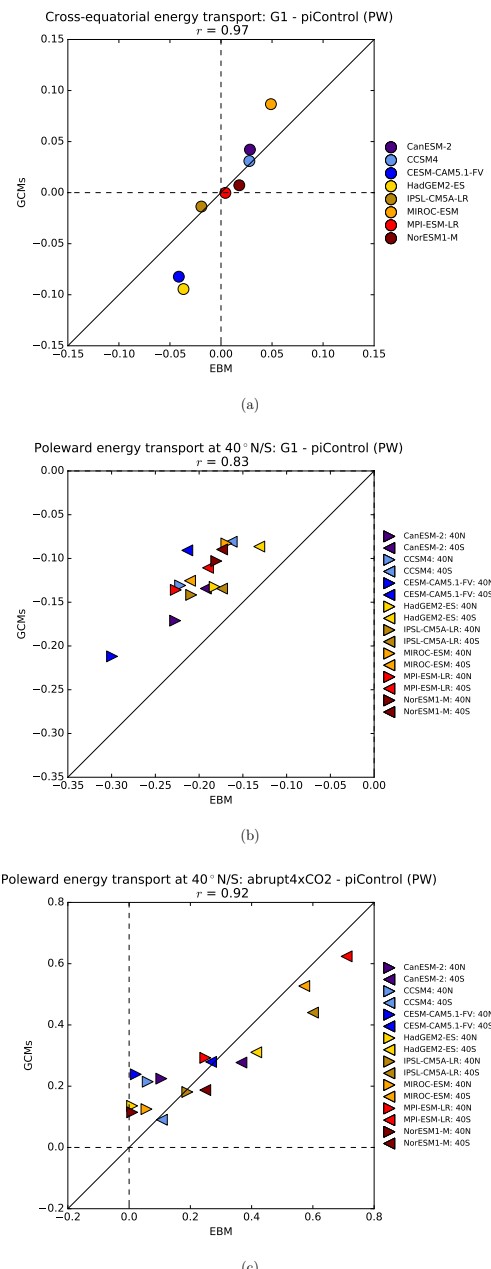

**Figure 5.** Meridional energy transport changes calculated by moist EBM ($x$-axis) versus those diagnosed from the GCM output ($y$-axis). (a): northward energy transport across the equator, for G1 minus piControl; (b): poleward energy transport changes across $40°$ N and S, for G1 minus piControl; (c): as in (b) but for abrupt4xCO2 minus piControl. Diagonal solid lines are 1:1 lines.

**Table 2.** Summary of attribution experiments run with moist energy balance model.

| Name | Effects considered | OLR fit from |
|------|--------------------|--------------|
| APRPcloud | SW radiative response at TOA due to cloud changes | piControl |
| APRPnoncloud | SW radiative response at TOA due to non-cloud atmosphere changes | piControl |
| APRPsurface | SW radiative response at TOA due to surface albedo changes | piControl |
| solarForcing | solar constant change (for G1), or nonlinear SW feedbacks (for abrupt4xCO2) | piControl |
| surfaceFlux | net SW, LW, sensible and latent heat flux changes at surface; *i.e.* surface energy storage | piControl |
| LWCRE | difference in net TOA LW radiation between clear-sky and all-sky conditions | piControl |
| greenhouse | enhanced greenhouse effect | G1 or abrupt4xCO2 |
| all_G1 | sum of effects listed in first 7 rows | G1 |
| all_4xCO2 | sum of effects listed in first 7 rows | abrupt4xCO2 |

The "OLR fit" refers to the fitting of coefficients for a linearized greenhouse effect based on surface air temperature and clear-sky OLR output from the GCMs; shown in the table is the GCM experiment from which these fits were drawn. Table 4 shows the actual fit coefficients.

changes more in the EBM than in the GCMs. Also, for abrupt4xCO2 - piControl, the EBM tends to underestimate poleward energy transport changes in the Northern Hemisphere and overestimate them in the Southern Hemisphere. With these cautions in mind regarding the exact magnitude of the changes, the EBM predicts changes in GCM-derived energy fluxes well enough to proceed to the attribution experiments.

## 3.2 Attribution of cross-equatorial energy transport changes

The attribution experiments are summarized in Table 2. Two experiments, "all_G1" and "all_4xCO2", consider all of the forcing and feedback terms for the two perturbation climates. There are three experiments that perturb shortwave feedbacks, based on the Approximate Partial Radiation Perturbation (APRP) method (Taylor et al., 2007). APRP uses a single-layer radiative transfer model to estimate the TOA radiative response to changes in clouds, non-cloud atmospheric scattering and absorption, and surface albedo, based on monthly mean GCM cloud fraction output and SW radiative flux output at the surface and TOA. We refer to our SW feedback attribution experiments as the "APRPcloud", "APRPnoncloud", and "APRPsurface" experiments, which consider cloud, non-cloud atmosphere, and surface albedo feedbacks, respectively. The "solarForcing" experiment considers the change in insolation at the TOA due to the solar constant change. The "surfaceFlux" experiment considers the change in the net downward energy flux at the surface, which represents energy storage and transport by the ocean. The "LWCRE" experiment considers changes in the LW cloud radiative effect, or the difference between clear-sky and all-sky net TOA LW fluxes. Finally, the "greenhouse" experiment considers the enhanced clear-sky greenhouse effect, which includes the $CO_2$ forcing and the water vapor, Planck and lapse rate feedbacks. The calculations of the input terms for the EBM for each experiment are described in more detail in Appendix B.

Figure 6 shows the changes in northward energy transport across the equator for G1 minus piControl in each of the experiments listed in Table 2. The "all_G1" results are the same as those plotted on the $x$-axis on Figure 5a and show that there is

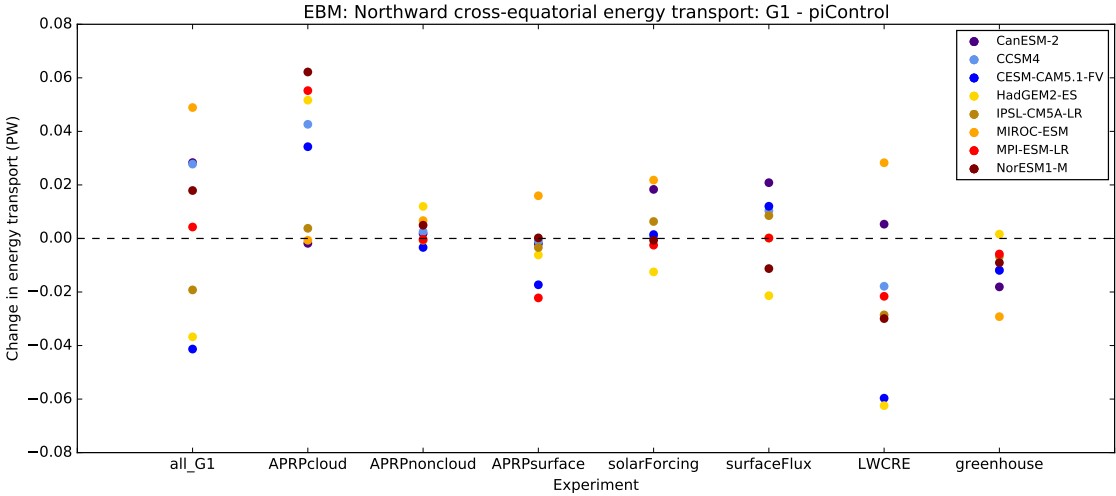

**Figure 6.** Changes in northward cross-equatorial energy transport calculated by moist EBM for G1 minus piControl in various attribution experiments.

considerable inter-model spread in the value of the cross-equatorial energy transport changes. None of the experiments shown in Figure 6 moves cross-equatorial energy transport in the same direction in all 8 models (although the APRPcloud and greenhouse experiments come close), so we cannot say with much confidence that any one forcing or feedback is likely to push the ITCZ one way or the other under a solar geoengineering scenario. However, it is useful to examine the inter-model spread in each experiment in order to determine which terms cause the most uncertainty in response of the ITCZ to solar geoengineering.

The two attribution experiments with the largest inter-model spread are the APRPcloud and LWCRE experiments, which correspond to SW and LW cloud feedbacks. This indicates that changes in clouds are the largest source of uncertainty regarding how cross-equatorial energy transport and, therefore, the ITCZ would respond to a hemispherically symmetric solar geoengineering scenario. This is similar to the finding of Frierson and Hwang (2012) that cloud feedbacks are the largest source of uncertainty for cross-equatorial energy transport changes in slab ocean simulations of $CO_2$-induced warming.

The APRPsurface, solarForcing, surfaceFlux, and greenhouse experiments have smaller inter-model spread than the two cloud experiments but are similar to each other. The spread in the surfaceFlux experiment indicates different responses of the atmosphere in different models to changes in either heat storage or cross-equatorial energy transport by the ocean. The appreciable inter-model spread in the solarForcing experiment suggests that the base state inter-hemispheric albedo difference is an important factor in the ITCZ response to solar geoengineering and solar forcings in general. This is interesting in the light of the result of Haywood et al. (2016) that tropical precipitation in the HadGEM2-ES is highly sensitive to the difference in the mean albedo between the hemispheres, and that equalizing them can improve GCM tropical precipitation biases.

## 3.3 Attribution of poleward energy transport changes

### 3.3.1 G1 minus piControl

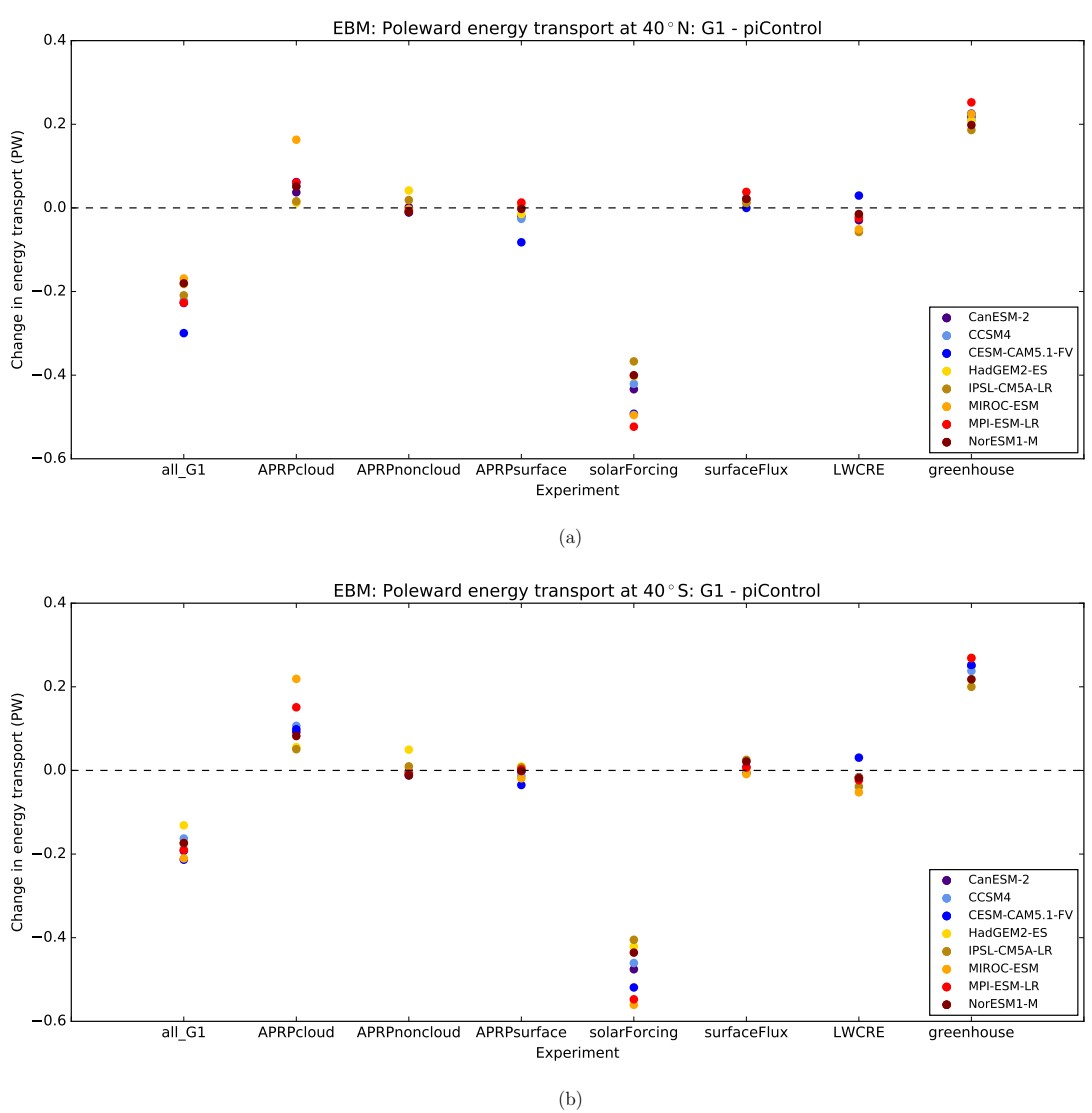

(a)

(b)

**Figure 7.** As in Figure 6 but for changes in poleward atmospheric energy transport across 40° N (a) and 40° S (b), for G1 minus piControl.

Figure 7 shows the results of the same attribution experiments shown in Figure 6, but for poleward energy transport at 40° N (Figure 7a) and 40° S (Figure 7b). In the all_G1 experiment, poleward energy transport decreases at this latitude in both hemispheres. Poleward energy transport decreases in the solarForcing experiment for each model, and increases in the

greenhouse experiment, but not by enough to compensate. The increase in poleward transport in the greenhouse experiment can be understood in terms of the increasing moisture transport argument discussed in Section 2 for the abrupt4xCO2 experiment. The $CO_2$ radiative forcing in the EBM is spatially uniform since OLR, in the initial perturbation, is reduced by the same amount everywhere (see Appendix B), but atmospheric moisture increases more in the tropics than at the poles because the atmosphere

was warmer in the tropics to begin with. The reduction in tropical moisture in the solarForcing case is greater than the increase in the greenhouse case because there is more sunlight to reduce in the tropics, causing a greater temperature perturbation there for solar reductions than for greenhouse gas increases. One caveat to this point is that in the actual atmosphere the $CO_2$ radiative forcing is stronger in the tropics than at the poles (although by as much as the solar forcing), which contributes to stronger poleward energy transport (Huang and Zhang, 2014); this mechanism for increased energy transport under greenhouse

gas forcings is not captured by the EBM.

The APRPcloud experiment exhibits an increase in poleward energy transport in both hemispheres in all models, which is consistent with a decrease in low cloud cover causing heating in the tropics. Schmidt et al. (2012) noted that low cloud cover decreased in four GCMs running G1. A more detailed investigation of the cloud changes in the full G1 ensemble and their physical mechanisms and radiative effects will be the subject of a future study. None of the other feedback experiments have a

consistent effect on poleward energy transport across 40° N/S, but the different feedback terms appear to rearrange the models in the all_G1 experiment, and contribute to the inter-model spread, with SW cloud feedbacks being the largest contributor. Models with a greater negative change in the solarForcing experiment (e.g. MPI-ESM-LR) also tend to have a greater positive change in the greenhouse experiment, and the compensation between these effects tends to reduce the inter-model spread. This implies that the remaining inter-model spread comes from the feedback terms. The fact that the solar forcing is the only term

contributing to the reduction of poleward energy transport in G1 in all models implies that the imperfect compensation between SW and LW forcings, not local feedbacks, causes this reduction.

### 3.3.2    abrupt4xCO2 minus piControl

Figure 8 is the equivalent of Figure 7, but for abrupt4xCO2 minus piControl. For abrupt4xCO2, the greenhouse attribution experiment results in an increase in poleward energy transport, similar to the same experiment for G1. SW cloud feedbacks

(APRPcloud experiment) are the largest contributor to the inter-model spread, followed by LW cloud feedbacks. Surface albedo feedbacks (APRPsurface) also contribute to the inter-model spread, but generally reduce poleward energy transport. (The increase in moisture due to tropical warming that results in greater poleward energy transport in the all_4xCO2 experiment does not show up when only surface albedo is perturbed.) This feedback term is mainly due to ice melt at high latitudes, which would be expected to reduce the equator-to-pole temperature gradient and therefore also reduce poleward energy transport. The

LWCRE experiment also reduces poleward energy transport for abrupt4xCO2, because the LW cloud feedback is positive at high latitudes due to an increase in the optical depth of high clouds (Zelinka et al., 2012). As with G1, non-cloud atmosphere SW feedbacks have small effects on the poleward energy transport for abrupt4xCO2, but there is a consistent increase in this case, presumably due to increases in SW absorption by water vapor. The solarForcing experiment in this case represents the

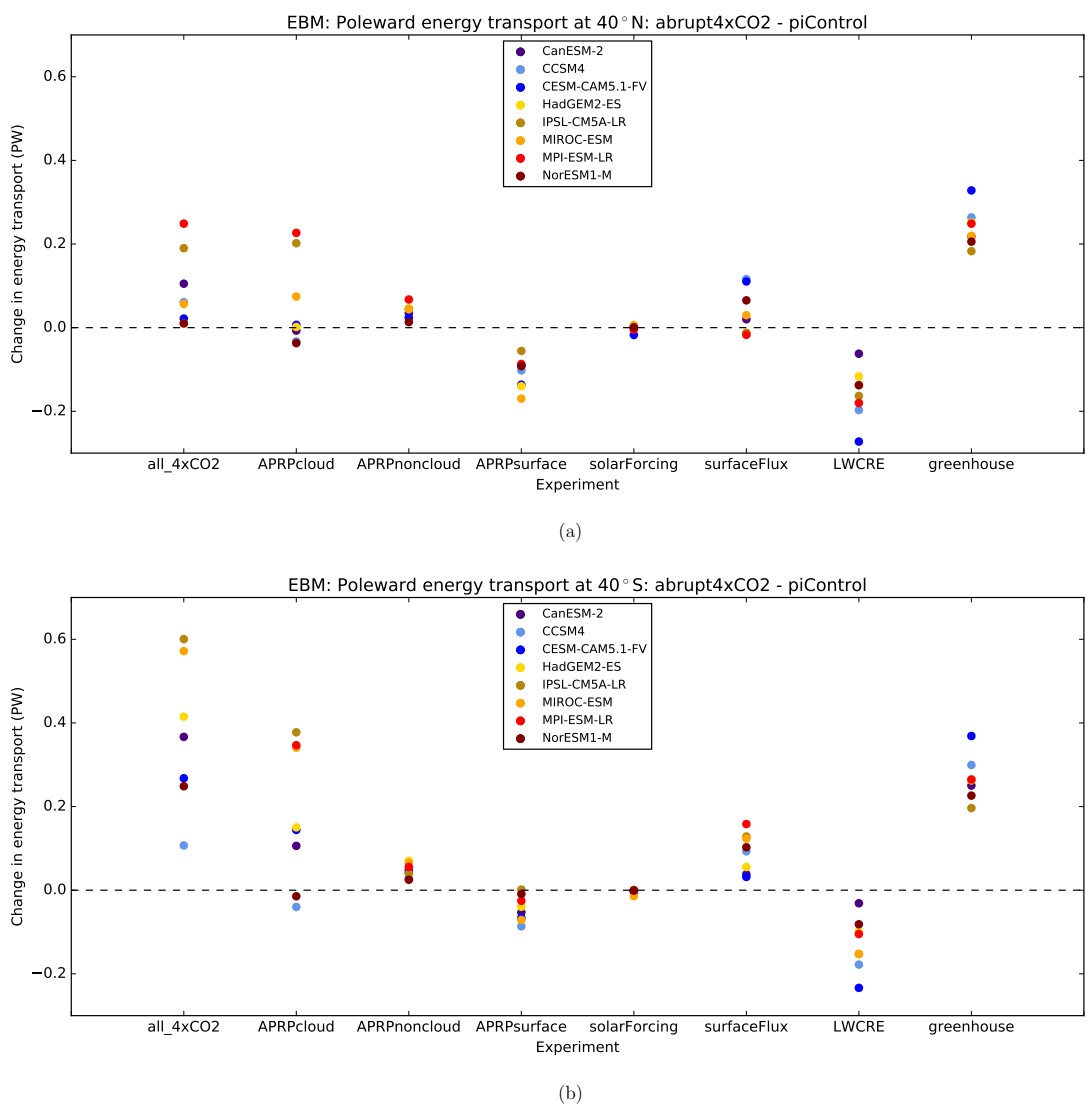

**Figure 8.** As in Figure 7 but for abrupt4xCO2 minus piControl.

residual between the total TOA net shortwave radiation changes and the individual feedback terms calculated using APRP; these effects are minor.

The surfaceFlux experiment has a much greater impact on poleward energy transport for abrupt4xCO2 than for G1, or for the 20th and 21st century CMIP3 runs analyzed by Hwang and Frierson (2010). This is because, while the TOA was kept approximately in energy balance in G1, and the imbalance is relatively small in 20th and 21st century runs, the abrupt4xCO2 case represents a response to an impulse that throws the climate system far out of equilibrium, with much energy being stored in

the oceans over time. The energy loss from the atmosphere to the ocean is strongest in high latitudes, leading to a compensating increase in poleward atmospheric energy transport in the abrupt4xCO2 surfaceFlux experiment.

## 4   Conclusions

Our analysis of the GeoMIP G1 ensemble shows that, when $CO_2$ concentrations are increased and the solar constant is reduced to compensate, poleward atmospheric energy transport decreases (Figures 1a,d). This is because of an increase in polar temperatures and decrease in tropical temperatures, or "residual polar amplification", that results from the different spatial patterns of the opposing solar and $CO_2$ forcings. The polar warming and tropical cooling cause a decrease in both dry static energy transport, which depends on the equator-to-pole temperature gradient, and latent heat transport, which depends on the meridional gradient of saturation vapor pressure. Residual polar amplification cannot be due to increases in poleward atmospheric energy transport, as might have been thought, because poleward energy transport actually decreases. It cannot be due to local feedbacks such as the ice-albedo feedback because these feedbacks cannot reverse the sign of an initial temperature change. Poleward energy transport by the ocean in the North Atlantic decreases (Hong et al., 2017), and there is no reason to expect an increase in poleward energy transport by the ocean overall given the decrease in net energy flux into the ocean in the tropics. Instead, the spatial distribution of the combined $CO_2$ and solar forcing causes this pattern of temperature change, while the decrease in poleward energy transport then acts as a negative feedback that limits the polar warming in G1.

The reduction of poleward energy transport helps explain why the difference in temperature change in the poles and the tropics is not nearly as much in G1 as it is in abrupt4xCO2, or in other words, why solar geoengineering in the form of a uniform solar constant reduction manages to eliminate most (but not all) of the polar amplification of $CO_2$-induced warming. The role of moisture transport is critical here. When $CO_2$ is increased by itself, poleward latent heat transport increases because of the large increase in moisture in the tropics, and this amplifies polar warming. In the G1 scenario, by contrast, the cooling of the tropics reduces the amount of moisture in the air, lessening the energy transport to the poles. This indicates that tropical moisture content is a very important control on the meridional temperature gradient. Geoengineering schemes have been designed that, in GCMs, avoid the problem of over-cooling the tropics by preferentially reducing sunlight in high latitudes (Ban-Weiss and Caldeira, 2010; Kravitz et al., 2016). It would be useful to analyze the changes in atmospheric energy transport in these scenarios in order to better understand the role moisture transport would play in attempting to regulate temperatures in various latitudes.

Our EBM attribution experiments illustrate the specific forcings and feedbacks responsible for the changes in meridional energy transport in G1. The solar forcing causes a reduction in poleward energy transport in mid-latitudes; the enhanced greenhouse effect only partially counteracts this. Cloud feedbacks are generally the largest contributors to the inter-model spread in both mid-latitude poleward energy transport changes and cross-equatorial energy transport changes, which are a predictor of ITCZ shifts. The large uncertainty in these quantities associated with clouds implies that an improved physical understanding of the changes in clouds in G1 would help our understanding of how regional precipitation and temperature

changes would play out under a solar geoengineering scenario. A more in-depth analysis of the cloud changes in the GeoMIP G1 ensemble will be the subject of a future study.

The finding that poleward atmospheric energy transport decreases in the G1 experiment relative to piControl is relevant for understanding why polar amplification of warming happens under increased $CO_2$. In warmer climates, the increased poleward energy transport contributes to the amount of polar amplification that occurs, as evidenced by model studies in which surface albedo is held constant (Alexeev et al., 2005; Graversen and Wang, 2009). However, our analysis of G1 shows that increases in poleward atmospheric energy transport are not necessary in order to have a decrease in the equator-to-pole temperature gradient. These results are particularly interesting in the light of the finding by Hwang et al. (2011) that polar amplification is actually negatively correlated with changes in atmospheric energy transport into the polar regions in CMIP3 global warming simulations. Our results reinforce the conclusion of Hwang et al. (2011) that changes in energy transport alone cannot predict changes in the meridional temperature gradient, which is actually governed by the coupling between energy transport and local feedbacks. It is useful to remember as well that radiative forcings are not spatially uniform, and the structure of the $CO_2$ forcing can affect atmospheric circulations in warming simulations (e.g. Huang and Zhang, 2014; Merlis, 2015). The spatial structure of radiative forcing is part of the set of processes, including local feedbacks and energy transport by the atmosphere and ocean, that interactively determine the Earth's meridional temperature pattern. Due to the complexity of these interactions, changes in the temperature gradient cannot be quantitatively predicted without a general circulation model. To better understand these interactions, it would be useful to do further analysis to quantify the contributions of different local feedbacks to the amount of polar warming in the G1 experiment.

## Appendix A: Details of GCM-derived energy transport calculations

Often, when calculating meridional energy transport based on a cumulative integration of energy flux convergence into the column from GCM output, there is a residual energy transport at the north pole, because the models' internal energy conservation involves terms that are not included in the reported fields energy flux fields, or else because the models used slightly different values of physical constants than we used in our diagnostics. For some models (CCSM4, CESM-CAM5.1-FV, GISS-E2-R, HadGEM2-ES, MIROC-ESM and NorESM1-M), this error can be reduced or nearly eliminated by adding $L_f P_{\text{snow}}$ to the right side of Equation 1, where $L_f$ is the latent heat of fusion of ice and $P_{\text{snow}}$ is the mass flux of snowfall at the surface. This term accounts for the net energy flux into the atmosphere when snow crystals form in the atmosphere and then melt on land or in the ocean. (We also do this for the calculation of moisture transport in Eq. 2.) For the rest of the models, including this term increases the north pole residual, so we omitted it, assuming that this term had been already accounted for inside the latent heat flux output field. Omitting this term in the first set of models as well did not significantly affect our results.

To correct for any remaining energy flux residual, we subtract the following error function $E$ from the northward energy transport profile:

$$E(\phi) = \frac{N}{2}\left(1 + \sin(\phi)\right) \tag{A1}$$

**Table 3.** Northward energy transport residual error at north pole in different models and runs, to 4 decimal places.

| Model | $N$ (piControl) (PW) | $N$ (G1 - piControl) (PW) | $N$ (abrupt4xCO2 - piControl) (PW) |
|---|---|---|---|
| CanESM-2 | -0.0531 | 0.0411 | 0.0809 |
| CCSM4 | 0.0162 | -0.0025 | 0.0015 |
| CESM-CAM5.1-FV | 0.0172 | -0.0138 | 0.0027 |
| CSIRO-Mk3L-1-2 | 0.1429 | -0.0093 | 0.0857 |
| GISS-E2-R | 0.0135 | -0.0002 | 0.0033 |
| HadGEM2-ES | -0.0270 | 0.0137 | 0.0164 |
| IPSL-CM5A-LR | 0.0373 | -0.0549 | 0.0027 |
| MIROC-ESM | -1.9135 | 0.0593 | 0.0734 |
| MPI-ESM-LR | -0.1174 | 0.0526 | -0.0515 |
| NorESM1 | 0.0137 | -0.00004 | 0.0027 |

where $\phi$ is the latitude and $N$ is the residual northward energy transport at the north pole. This correction function assumes that each unit area of Earth's surface contributes equally to the error. To demonstrate that the error is small, Table 3 shows the values of $N$ in piControl and the change in $N$ in the other 2 runs relative to piControl. The errors are generally small (< .15 PW), except for MIROC-ESM, but even in this case the difference in the error between the runs is still small (all models have error < .06 PW for G1 minus piControl, or .09 PW for abrupt4xCO2 minus piControl). Once the correction in Eq. (A1) is applied, the energy transport residual should only affect the results (in terms of differences between runs) if the errors are spatially nonuniform and the spatial pattern of the error differs between the runs. Since even the total error differences are small between runs, these residuals should not be a significant source of error in our analysis.

## Appendix B: Details of moist EBM calculations

We use the moist energy balance model first used in Hwang and Frierson (2010). Here we describe how the model works, with an emphasis on new changes made for the solar geoengineering experiments.

The core equation of the model, as in other energy balance models (e.g. North, 1975), is a heat diffusion equation:

$$\frac{\partial T_s}{\partial t} = C \left( \text{EI} - \text{OLR} + \frac{p_s}{g} D \nabla^2 \text{MSE} \right) \tag{B1}$$

or

$$\frac{\partial T_s}{\partial t} = C \left( \text{EI} - \text{OLR} + \frac{p_s}{g} \frac{D}{r^2} \frac{\partial}{\partial x} \left[ (1 - x^2) \frac{\partial \text{MSE}}{\partial x} \right] \right), \tag{B2}$$

where MSE is the moist static energy, $T_s$ is surface temperature, $C$ is an arbitrary surface heat capacity, OLR is the outgoing longwave radiation at the top of the atmosphere, EI ("energy in") is the net surface and TOA energy flux into the atmospheric

column excluding OLR, $D$ is a diffusivity coefficient for MSE, $p_s$ is the surface pressure, $g$ is the acceleration due to gravity, $r$ is the radius of the earth, and $x = \sin\theta$ where $\theta$ is the latitude. We have explicitly written out the $r^2$ that comes from the Laplacian operator in equation B2 rather than absorbing it into $D$ as is often done (e.g. North, 1975). Noting that $\mathrm{d}x = \cos\theta\,\mathrm{d}\theta$, equation B2 can also be written in terms of latitude, which is more convenient in terms of specifying inputs for EI as functions

of latitude without converting to sine latitude first:

$$\frac{\partial T_s}{\partial t} = C\left(\mathrm{EI} - \mathrm{OLR} + \frac{p_s}{g}\frac{D}{r^2}\frac{\partial}{\cos\theta\,\partial\theta}\left[\cos\theta\frac{\partial\mathrm{MSE}}{\partial\theta}\right]\right). \tag{B3}$$

We assume a value of $1.06 \times 10^6$ m$^2$ s$^{-1}$ for $D$, following Hwang and Frierson (2010), and a flat topography with $g$ = 9.8 m s$^{-2}$ and $p_s$ = 980 hPa. We step forward in time with a relative time step of $\frac{dt}{C} = 1 \times 10^{-4}$. The model is considered to have converged when $T_s$ differs by less than .001 K everywhere in the domain between successive time steps.

The moist static energy is calculated according to:

$$\mathrm{MSE} = C_p T_s + L_v q \tag{B4}$$

where $C_p$ is the heat capacity of air at constant pressure, $L_v$ is the latent heat of vaporization of water, and $q$ is the specific humidity. We calculate $q$ as a function of $T_s$ using Equation 3, assuming a relative humidity of 80%.

The OLR is treated as a linear function of temperature:

$$\mathrm{OLR} = aT_s - b \tag{B5}$$

where the coefficients $a$ and $b$ are calculated as linear least-squares fits from the monthly surface air temperature and clear-sky OLR output in each of the GCMs over the first 40 years of piControl. To consider the enhanced greenhouse effect in "perturbation" climates (G1 and abrupt4xCO2), we fit new coefficients $b'$, maintaining the original value of $a$ (following Hwang and Frierson (2010)), based on the surface temperature and clear-sky OLR output in those experiments. Table 4 shows

the values of $a$, $b$, and $b'$ we calculated for each of the GCMs.

To run the EBM, we input the $a$ and $b$ coefficients shown in Table 4, and an EI term calculated differently for the different attribution experiments. For the EBM runs representing piControl conditions, we combine the following terms from the zonal mean output of each GCM:

$$\mathrm{EI}_{\mathrm{piControl}} = S - L_C + F_s \tag{B6}$$

where $S$ is the net downward SW radiation at the TOA, $L_C$ is the LW cloud radiative effect (clear-sky OLR minus all-sky OLR), and $F_s$ is the net upward surface flux, including SW and LW radiation, sensible heat flux and latent heat flux.

For the "full" perturbation runs emulating the G1 and abrupt4xCO2 experiments, we use $b'$ instead of $b$ for the OLR calculation, and the EI term is:

$$\mathrm{EI}_{\mathrm{perturb}} = \mathrm{EI}_{\mathrm{piControl}} + C_S + A_s + I + \Delta L_C + O + \Delta S \tag{B7}$$

**Table 4.** Values of fit coefficients for clear-sky OLR as a function of temperature for use in moist EBM analysis.

| Model | $a$ (W m$^{-2}$ K$^{-1}$) | $b$ (W m$^{-2}$) | $b'$ (G1) | $b'$ (abrupt4xCO2) |
|---|---|---|---|---|
| CanESM-2 | 2.0667 | 326.83 | 334.99 | 335.00 |
| CCSM4 | 2.1604 | 350.06 | 358.31 | 360.35 |
| CESM-CAM5.1-FV | 2.0724 | 328.98 | 337.74 | 341.62 |
| HadGEM2-ES | 2.1531 | 349.37 | 357.38 | 358.99 |
| IPSL-CM5A-LR | 2.2149 | 363.39 | 370.58 | 370.46 |
| MIROC-ESM | 2.0512 | 327.40 | 336.37 | 336.18 |
| MPI-ESM-LR | 2.0157 | 315.55 | 324.47 | 324.34 |
| NorESM1 | 2.1403 | 346.36 | 354.38 | 354.68 |

where $C_S$, $A_s$ and $I$ are the change in the net downward TOA SW radiation associated with cloud, non-cloud atmosphere, and surface albedo feedbacks, respectively, calculated using the Approximate Partial Radiation Perturbation (APRP) method (Taylor et al., 2007); $\Delta L_C$ is the change in the LW cloud radiative effect in the GCM output; $O$ is the change in the net surface flux in the GCM output; and $\Delta S$ is the change in the solar forcing. We calculate $\Delta S$ by taking the change in net TOA SW radiation between the control and perturbation climates, and subtracting $C_S$, $A_s$, and $I$ to get the change in solar radiation that is not due to any of the three feedback terms. In G1 this represents the effect of changing the solar constant; in abrupt4xCO2 this represents a residual feedback not accounted for by a linear sum of the other 3 feedbacks.

For the individual attribution experiments (except "greenhouse"), we use $b$ in the OLR calculation, and the EI terms are calculated as follows (experiment labels following Table 2):

$$\text{EI}_{\text{APRPcloud}} = \text{EI}_{\text{piControl}} + C_S \tag{B8}$$

$$\text{EI}_{\text{APRPnoncloud}} = \text{EI}_{\text{piControl}} + A_s \tag{B9}$$

$$\text{EI}_{\text{APRPsurface}} = \text{EI}_{\text{piControl}} + I \tag{B10}$$

$$\text{EI}_{\text{solarForcing}} = \text{EI}_{\text{piControl}} + \Delta S \tag{B11}$$

$$\text{EI}_{\text{surfaceFlux}} = \text{EI}_{\text{piControl}} + O \tag{B12}$$

$$\text{EI}_{\text{LWCRE}} = \text{EI}_{\text{piControl}} + \Delta L_C \tag{B13}$$

For the "greenhouse" experiment, we use the control value of EI, but use $b'$ instead of $b$ for the OLR calculation.

The northward energy transport output by the EBM and shown in figures 6 through 8 is the cumulative meridional integration of the MSE diffusion term in Eq. (B2). The discretization of the diffusion equation for numerical solving inevitably results in some loss of energy, so after integrating, we apply a correction for residual northward transport at the North Pole for the EBM results, using Eq. (A1).

*Author contributions.* R.D. Russotto analyzed the GCM output, ran the EBM attribution experiments, produced the figures, and wrote the bulk of the paper. T.P. Ackerman provided general guidance and assisted with the preparation of the manuscript text.

*Code availability* All scripts used to analyze data and create plots are available here: atmos.washington.edu/articles/GeoMIP_EnergyTransport_PolarAmplification_ITCZ. A standalone Python code for the APRP method is
available at: https://github.com/rdrussotto/pyAPRP.

*Acknowledgements.* This work was supported by the U.S. Department of Defense (DoD) through the National Defense Science and Engineering Graduate Fellowship (NDSEG) Program and by a grant to JISAO from the Fund for Innovative Climate and Energy Research. We thank Dargan Frierson for providing helpful guidance regarding the direction of this research, and Kyle Armour for pointing out a possible connection between decreased poleward moisture transport and reductions in mid-latitude precipitation. For their roles in producing, coordi-
nating, and making available the CMIP5 and GeoMIP model output, we acknowledge the climate modeling groups (listed in Table 1 of this paper), the World Climate Research Programme's (WCRP) Working Group on Coupled Modelling (WGCM), and the Global Organization for Earth System Science Portals (GO-ESSP). We are grateful to Ben Kravitz, Helene Muri, Ulrike Niemeier, Stephen Phipps, and Jin-Ho Yoon for helping to provide access to GeoMIP output that was not yet available online; Phil Rasch, Balwinder Singh, and Ashly Spevacek for helpful discussions on technical issues with energy budgets in CMIP5 models; and Yen-Ting Hwang for providing us with codes for
the APRP method and the moist EBM that formed the basis of this analysis. Two anonymous reviewers provided comments that greatly helped to improve the manuscript. We are particularly grateful to Reviewer 2 for their detailed commentary on the methodology of our EBM experiments. This led to us finding and fixing a major error in our implementation of the EBM.

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
