# Peer review of "Energy transport, polar amplification, and ITCZ shifts in the GeoMIP G1 ensemble"

_Atmospheric Chemistry and Physics, 2017_

## Referee Comment (RC1) · Anonymous Referee #1 · 28 Sep 2017

**Review of "Energy transport, polar amplification, and ITCZ shifts in the GeoMIP G1 ensemble" by R. Russotto and T. Ackerman (ACPD, 2017)**

**General Comments**

In this study the authors investigate the impacts of solar geoengineering and global warming on atmospheric meridional energy fluxes by using results from an intermodel ensemble (GeoMIP) and by employing a moist energy balance model (EBM). The EBM is combined with GeoMIP output in order to attribute energy transport changes to different climate forcing agents and climate feedbacks. The study seeks to answer two separate questions related to meridional energy transport: 1. how does solar geo impact the poleward energy transport at high latitudes, and 2. how does solar geo impact the cross-equatorial energy flux. In essence, the first question asks whether solar geo would exacerbate or counteract high-latitude warming while the second question asks which aspects of GCMs contribute most to uncertainty in changes to tropical precipitation. The authors show that solar geo would counteract the poleward energy transport enhancement under global warming, which they rather elegantly attribute to changes in moisture transport. This result helps to explain the observed mid-latitude drying signal in previous solar geo studies (e.g. Tilmes et al., 2013) and is a very useful inference. It also helps to explain why solar geo effectively (though not completely) offsets high-latitude warming. In an attribution study with an EBM, the authors attribute the largest source of uncertainty in poleward energy transport changes to cloud feedbacks, in agreement with previous studies looking at energy transport under global warming. They also show that cloud feedbacks contribute most to uncertainties in cross equatorial energy transfer (and hence tropical rainfall migration) under solar geo.

The paper is a very useful contribution to the solar geoengineering literature and helps to shed light on important results of previous studies such as residual high-latitude warming. The background information (section 1) is comprehensive and the methodology is sound. The use of an EBM is appropriate, although I think the disparities between the results of the EBM and GCMs (e.g. Fig. 5) should be elucidated more carefully in the text. My primary concern with the paper is the lack of consideration for oceanic energy transport changes – in particular for cross-equatorial energy fluxes where oceanic energy transport is important. The manuscript would benefit from looking at oceanic energy transport changes explicitly (a methodology for calculating meridional oceanic energy transport is provided by Hawcroft et al, 2016). This may elucidate why the ITCZ shifts in abrupt4xco2 are not correlated with atmospheric cross-equatorial energy transport changes. If the authors are unable or unwilling to investigate oceanic energy fluxes, then I would suggest altering the title of the manscript to "*Atmospheric* energy transport, polar amplification, and ITCZ shifts in the GeoMIP G1 ensemble" to better reflect the paper's contribution. Once these changes (and various minor changes listed below) are made, I'd be happy to recommend publication.

**Specific Comments**

| P1 L5: | First instance of $CO_2$ – define as carbon dioxide |
|---|---|
| P1 L8: | Mention explicitly that it is the *radiative forcing* from enhanced GHGs that is being offset by solar reduction in G1 |
| P1 L8: | Sentence beginning "In G1,..." - consider starting with "We show that ..." to distinguish your results from prior results |
| P1 L19: | Consider replacing "compensated for" with "counteracted". Also add a suitable reference at the end of this sentence |
| P1 L23: | Sentence beginning "since reflecting sunlight would affect ..." is ambiguous. Explain why solar geoengineering leaves residual warming at high latitudes – at least give the primary theories – e.g. more sunlight in the tropics on average – with a suitable reference (e.g. Kravitz et al., 2013) |

P2 L12: Replace "subtropical" with "tropical" - Haywood et al identified Sahelian drought as a concern of hemispheric geoengineering and did not look at the subtropics

P2 L14: Add a suitable reference to the last sentence of this paragraph – which study explicitly identifies an ITCZ shift following a symmetric SAI application?

P2 L15: You say "ITCZ shifts are closely related to the meridional transport of energy by the atmosphere". This is irrefutable, although you should add an appropriate reference, but only tells half the story. Add more discussion about the relative importance of ocean heat transport here, and its ability to control ITCZ position. The following references may be useful: Green and Marshall (2017), Hawcroft et al (2016), Haywood et al (2016), Hwang et al (2017), Marshall et al (2014)

P2 L30: The aim of G1 is not to "keep global mean temperature at approximately preindustrial levels" as you say – be more explicit about the simulation design

P4 L5: Define $\nabla \cdot F_L$ at first use (i.e. latent energy transport)

Figs 1 and 2: Consider also changing the y-units to 'poleward energy transport' rather then 'northward energy transport' to assist comparisons. Lastly, I'd suggest putting 'Latitude (N)' as the x-title rather than 'Latitude'

P7 L2: Add a suitable reference for the DSE response to high-latitude warming

Eqn 3: This equation is valid for temperature in units in Kelvin, whereas your plots give temperature in units of oC – pick one for consistency and use throughout the manuscript – I'd personally go with the SI unit K

P8 L14: Sentence "This leaves the differing spatial patterns of forcings as the only possible explanation" should have a caveat that meridional ocean heat transfer is negligible at high-latitudes (e.g. Fig. 6 in Hawcroft et al 2016)

**P8 L30:** I recommend that the authors explicitly look at changes to meridional ocean heat transfer in the G1 and abrupt4XCO2 simulations using the methodology of Hawcroft et al (2016). Whilst the caveat about ocean heat transport in P8 L30 is appreciated, explicity looking at changes to ocean heat content / transport would significantly improve the manuscript whilst not altering the primary results

Fig. 4: Consider adding the correlation coefficients to the the respective figures

P9 L2: Reference the Haywood et al (2013) study at the end of "shifts toward the warmed hemisphere"

P9 L2: Change "It implies" to "The ITCZ shifts in Fig 4b imply"

P10 L10: Change "40 $^{\circ}$N" to "40 $^{\circ}$N/S"

P10 L11: Add a space between piControl and (Figure 5c)

**P10 L12:** Yes there are strong correlations, but that doesn't mean the EBM is doing a good job! For instance, for abrupt4XCO2 the EBM predicts a negative poleward energy transport anomaly at 40 $^{\circ}$N for 6 out of the 8 models where the GCMs give a positive anomaly. This issue should be discussed and not glossed over.

Fig. 5: A minor suggestion - put "40 $^{\circ}$S/N" into the plot titles for b) and c)

Fig. 6: In the caption and the titles note that it is the *northward* energy transport that you are plotting

P13 L5: Again – I urge you to explicitly assess oceanic heat uptake in these simulations – you say that the results of the EBM imply that oceanic heat uptake differs between the GCMs – it would not be difficult to assess this hypothesis and would add value to your results

P13 L7: Sentence beginning "The impact of the solar forcing term ..." is very wordy and does not read well – rephrase

P13 L25: "has" → "have"

P13 L27: Sentence beginning "Models with a greater negative change..." - include an example

P13 L30: I would also use Fig. 6 in Hawcroft et al (2016) to argue your point that poleward energy transport changes in are likely not due to oceanic heat transport changes – i.e. more background meridional heat transport in the atmosphere than the ocean

P14 L8:      You should add caveats that the EBM does not get the sign of the energy transport right in the Northern Hemisphere (Fig. 5c). I would seriously consider removing Fig. 8A and associated analysis due to this issue as it indicates the EBM is missing the point and any analysis is compromised
P16 L18:     Following "poleward atmospheric energy transport decreases" refer to Figs 1a,d

**References**

Green, B., and J. Marshall (2017), Coupling of Trade Winds with Ocean Circulation Damps ITCZ Shifts, J. Clim., 30(12), 4395–4411, doi:10.1175/JCLI-D-16-0818.1

Hawcroft, M., J. M. Haywood, M. Collins, A. Jones, A. C Jones, and G. Stephens (2016), Southern Ocean albedo, inter-hemispheric energy transports and the double ITCZ: Global impacts of biases in a coupled model, Clim. Dyn., 1–17, doi:10.1007/s00382-016-3205-5.

Haywood, J. M., et al. (2016), The impact of equilibrating hemispheric albedos on tropical performance in the HadGEM2-ES coupled climate model, Geophys. Res. Lett., 43, 395–403, doi:10.1002/2015GL066903.

Hwang, Y.-T., S.-P. Xie, C. Deser, and S. M. Kang (2017), Connecting tropical climate change with Southern Ocean heat uptake, Geophys. Res. Lett., 44, doi:10.1002/2017GL074972.

Marshall, J. A., A. Donohoe, D. Ferreira, and D. McGee, 2014: The ocean's role in setting the mean position of the inter-tropical convergence zone. Climate Dyn., 42, 1967–1979, doi:10.1007/s00382-013-1767-z.

Tilmes, S., Fasullo, J., Lamarque, J.-F., Marsh, D. R., Mills, M., Alterskjær, K., Muri, H., Kristjánsson, J. E., Boucher, O., Schulz, M., Cole, J. N. S., Curry, C. L., Jones, A., Haywood, J., Irvine, P. J., Ji, D., Moore, J. C., Karam, D. B., Kravitz, B., Rasch, P. J., Singh, B., Yoon, J.-H., Niemeier, U., Schmidt, H., Robock, A., Yang, S., and Watanabe, S.: The hydrological impact of geoengineering in the Geoengineering Model Intercomparison Project (GeoMIP), Journal of Geophysical Research: Atmospheres, 118, 11,036–11,058, doi:10.1002/jgrd.50868, 2013.

---

## Referee Comment (RC2) · Anonymous Referee #2 · 5 Oct 2017

**Overview**

Overall, this manuscript makes an interesting contribution to the physical climate's response to solar radiation management geoengineering. The analysis framework is centered on moist static energy transport, which has been a valuable way of understanding other climate perturbation (e.g., from carbon dioxide and other anthropogenic radiative forcing). Bringing this perspective to the geoengineering simulations is helpful to understand the ways in which the reduced solar constant is not perfectly offsetting the regional climate changes from increased carbon dioxide, even if the global-mean surface temperature is near zero. The conclusions concerning the critical role of the spatial pattern of the forcing (vs. the possibility that radiative feedbacks differ between solar and greenhouse gas forcing) is nice to broadcast clearly, as the abstract does.

I think the manuscript is suitable for publication, but there are some important caveats about the analysis that need to be added and the authors need to alter how they perturb the moist EBM to be more consistent with what we know about the radiative forcing of carbon dioxide (at least the global mean part).

**Major Points**

**1) ITCZ shifts: recent literature**

There are a few relevant new publications that the authors should engage with.

Seo et al. (2017) showed that the cross-equatorial energy flux does not always account for ITCZ shifts in GCM simulations of carbon dioxide warming. It's fine to continue to use this as a first cut toward understanding the simulations' behavior, but its limitations need to be acknowledged. It may be that the non-zero intercept in Fig. 4a comes about for a reason discussed there: the gross moist stability (energetic stratification) can also change.

Viale and Merlis (2017) analyzed simulations with solar forcing and carbon dioxide forcing and found the ITCZ shifted a different amount. They interpreted this result through the refined energy flux equator argument described by Bischoff and Schneider (2014). The result in Viale and Merlis (2017) is in line with the typical GCM results here: more sensitivity to carbon dioxide than solar forcing in ITCZ shifts, so a poleward shift in geoengineered climate. This comes about by a different spatial pattern in the radiative forcing (not differences in radiative feedbacks).

Last, the TRACMIP simulations (Voigt et al., 2016) have examples of radiatively forced warming where the ITCZ shift does not follow the energy flux equator. I've seen this shown in conference talks, though it may not have been published yet.

I think this literature needs to be invoked rather than ocean heat storage (p. 8 L30) for the breakdown in abrupt 4x simulations: the ocean heat storage is already taken into account through the surface fluxes. That cannot explain the why the cross equatorial energy fluxes in the atmosphere are uncorrelated to the ITCZ shift.

**2) Moist EBM analysis**

First, I appreciate that the methodological details are described in a greater detail than in previous papers that use this approach and the authors have made their codes available.

* Eqn. B1 has a mistake (see North 1975 and subsequent diffusive EBM papers). There is a missing $(1 - x^2)$:

$$\frac{\partial}{\partial x}\left[(1 - x^2)\frac{\partial MSE}{\partial x}\right].\tag{1}$$

I hope this is a typesetting mistake and not a problem with the implementation of the EBM used for the analysis.

* The convention of $OLR = aT_s - b$ is confusing when other EBM literature has the opposite use of $a$ and $b$ (e.g., $OLR = a + bT_s$), but I understand the desire for continuity with Hwang and Frierson (2010).

* I think it's valuable to try to set the record straight concerning how Hwang and Frierson (2010) described the change they made to do the warming EBM simulations. I thought they made a terminology mistake, saying that they re-fit $b$ "to capture the change in climate sensitivity", which would be a change in $a$. Changing just $b$ is reasonable as you can think of that as a change in radiative forcing (e.g., Rose et al., 2014). This manuscript, however, suggests both $a$ and $b$ must be changed to simulate altered climates. I don't follow that, as Rose et al. (2014) and others have done moist EBM simulations that are just changing $b$ and have changes in energy transport. The values of $a$ don't seem to change too much, generally near 2 W/m$^2$/K as expected from the sum of temperature and water vapor feedbacks, so I'm confused.

* Again, I want to praise the authors for disclosing the methodological details. I have not seen the parameter values for $a$ and $b$ when using linear fits to GCMs before (Table 4). However, the results are troubling and perhaps earlier work would have attracted scrutiny if this had been disclosed. If we think of $b$ as representing the carbon dioxide radiative forcing (change in OLR with unchanged surface temperature), the fits imply carbon dioxide radiative forcing that varies across models from 0 to 16 W/m$^2$ (just considering the abrupt 4x column vs. the control). This is a problem! It also affects the G1 simulations.

* The last important thing that should be noted is that the approach of prescribing TOA perturbations (e.g., from cloud radiative effect changes) in the moist EBM does not capture the fact that feedbacks are interacting with each other (Feldl et al., 2017) and with the changing energy transport (Rose et al., 2014; Merlis, 2014; Rose and Rayborn, 2016). There's still value in this way of diagnosing things (partially interactive), but it's not the full story and may help explain why EBM is changes less than the GCM.

**3) Forcing estimates**

The authors describe in conclusions that the radiative forcing for carbon dioxide is not uniform, but this seems important enough to come earlier when describing the EBM set up (p. 10 L1).

* The authors cite only Zelinka and Hartmann (2012) (p. 15) when discussing the extratropical response to carbon dioxide. Those authors neglected the structure of carbon dioxide, which was subsequently included in Huang and Zhang (2014). They also looked at the same CMIP5 abrupt 4x simulations, so it's important to confirm consistency.

* Other authors (e.g., Feldl and Bordoni, 2016) have taken the approach of diagnosing the forcing, including its spatial structure, from fixed SST simulations ("troposphere adjusted forcing"). Given my concern about the linear fit estimate of $b$ values above, I think this is a superior way to drive the moist EBM. Another alternative would be to take previously published global-mean adjusted forcing estimates and use those—at least all the carbon dioxide forcing would be near 7 W/m$^2$.

**Minor Points**

p. 3 L19: "moisture fluxes" terminology is not ideal for the latent surface fluxes, given that meridional energy and moisture fluxes are discussed later.

Fig. 3: how close is the global-mean surface temperature change to zero in these simulations? I understand this is discussed in other papers using G1, but it would be helpful to at least state a representative number (e.g., within 0.2 K).

Fig. 3 caption: presumably these are zonal *and annual* mean *surface air* temperature changes? I didn't see this in the main text either. Likewise for saturation vapor pressure.

p. 8 L27, and elsewhere: a big excessive on the number of digits for $r$.

Fig. 5: plot 1-to-1 lines here

p. 16 L6: the ocean heat uptake is weighted toward subpolar oceans (vs. a uniform uptake which wouldn't affect extratropical atmospheric energy transport)

**References**

Bischoff, T. and T. Schneider, 2014: Energetic constraints on the position of the Intertropical Convergence Zone. *J. Climate*, **27**, 4937–4951.

Feldl, N. and S. Bordoni, 2016: Characterizing the Hadley circulation response through regional climate feedbacks. *J. Climate*, **29**, 613–622.

Feldl, N., S. Bordoni, and T. M. Merlis, 2017: Coupled high-latitude climate feedbacks and their impact on atmospheric heat transport. *J. Climate*, **30**, 189–201.

Huang, Y. and M. Zhang, 2014: The implication of radiative forcing for poleward energy transport. *Geophys. Res. Lett.*, **41**, 1665–1672.

Hwang, Y.-T. and D. M. W. Frierson, 2010: Increasing atmospheric poleward energy transport with global warming. *Geophys. Res. Lett.*, **37**.

Merlis, T. M., 2014: Interacting components of the top-of-atmosphere energy balance affect changes in regional surface temperature. *Geophys. Res. Lett.*, **41**, 7291–7297.

Rose, B. E. J., K. C. Armour, D. S. Battisti, N. Feldl, and D. D. B. Koll, 2014: The dependence of transient climate sensitivity and radiative feedbacks on the spatial pattern of ocean heat uptake. *Geophys. Res. Lett.*, **41**, 1071–1078.

Rose, B. E. J. and L. Rayborn, 2016: The effects of ocean heat uptake on transient climate sensitivity. *Current Climate Change Reports*, **2 (4)**, 190–201.

Seo, J., S. Kang, and T. M. Merlis, 2017: A model intercomparison of the tropical precipitation response to a $CO_2$ doubling in aquaplanet simulations. *Geophys. Res. Lett.*, **44**, 993–1000.

Viale, F. and T. M. Merlis, 2017: Variations in tropical cyclone frequency response to solar and $CO_2$ forcing in aquaplanet simulations. *J. Adv. Model. Earth Syst.*, **9**, 4–18.

Voigt, A., et al., 2016: The tropical rain belts with an annual cycle and a continent model intercomparison project: TRACMIP. *J. Adv. Model. Earth Syst.*, **8**, 1868–1891.

Zelinka, M. D. and D. L. Hartmann, 2012: Climate feedbacks and their implications for poleward energy flux changes in a warming climate. *J. Climate*, **25**, 608–624.

---

## Author Comment (AC1) · 20 Dec 2017

**Responses to Reviewer Comments**

**Energy Transport, Polar Amplification, and ITCZ Shifts in the GeoMIP G1 Ensemble**

**R. D. Russotto and T. P. Ackerman**

**Responses to reviewer comments in blue. Following the responses are a revised manuscript showing differences from the original version, and then a clean revised manuscript.**

**Reviewer 1**

**General Comments**

In this study the authors investigate the impacts of solar geoengineering and global warming on atmospheric meridional energy fluxes by using results from an intermodel ensemble (GeoMIP) and by employing a moist energy balance model (EBM). The EBM is combined with GeoMIP output in order to attribute energy transport changes to different climate forcing agents and climate feedbacks. The study seeks to answer two separate questions related to meridional energy transport: 1. how does solar geo impact the poleward energy transport at high latitudes, and 2. how does solar geo impact the cross-equatorial energy flux. In essence, the first question asks whether solar geo would exacerbate or counteract high-latitude warming while the second question asks which aspects of GCMs contribute most to uncertainty in changes to tropical precipitation. The authors show that solar geo would counteract the poleward energy transport enhancement under global warming, which they rather elegantly attribute to changes in moisture transport. This result helps to explain the observed mid-latitude drying signal in previous solar geo studies (e.g. Tilmes et al., 2013) and is a very useful inference. It also helps to explain why solar geo effectively (though not completely) offsets high-latitude warming. In an attribution study with an EBM, the authors attribute the largest source of uncertainty in poleward energy transport changes to cloud feedbacks, in agreement with previous studies looking at energy transport under global warming. They also show that cloud feedbacks contribute most to uncertainties in cross equatorial energy transfer (and hence tropical rainfall migration) under solar geo.

The paper is a very useful contribution to the solar geoengineering literature and helps to shed light on important results of previous studies such as residual high-latitude warming. The background information (section 1) is comprehensive and the methodology is sound. The use of an EBM is appropriate, although I think the disparities between the results of the EBM and GCMs (e.g. Fig. 5) should be elucidated more carefully in the text. My primary concern with the paper is the lack of consideration for oceanic energy transport changes – in particular for cross-equatorial energy fluxes where oceanic energy transport is important. The manuscript would benefit from looking at oceanic energy transport changes explicitly (a methodology for

calculating meridional oceanic energy transport is provided by Hawcroft et al, 2016). This may elucidate why the ITCZ shifts in abrupt4xco2 are not correlated with atmospheric cross-equatorial energy transport changes. If the authors are unable or unwilling to investigate oceanic energy fluxes, then I would suggest altering the title of the manscript to "*Atmospheric* energy transport, polar amplification, and ITCZ shifts in the GeoMIP G1 ensemble" to better reflect the paper's contribution. Once these changes (and various minor changes listed below) are made, I'd be happy to recommend publication.

Thank you for the detailed suggestions in this review. The differences between the EBM and GCM results are smaller now that a major bug has been fixed in our implementation of the EBM (see response to Reviewer 2's Major Point 2). We add more discussion of the EBM-GCM differences in this version.

As for ocean transport, we have added some discussion of this to the introduction, However, we still think our focus on atmospheric energy transport makes sense. As Reviewer 2 points out, any changes in oceanic heat transport or storage are already taken into account in the atmospheric energy budget via the surface fluxes.

The *poleward* ocean heat transport is important to another aspect of this study: the explanation for the residual polar amplification in G1. We originally did not adequately discuss the possibility of an increase in ocean heat transport as a possible reason for this, but Hong et al. (2017) found that poleward heat transport by the ocean (northward across 30 degrees N in the Atlantic) decreases in G1, like that by the atmosphere. We now discuss this in the paper.

**Specific Comments**

P1 L5: First instance of CO2 – define as carbon dioxide

Done.

P1 L8: Mention explicitly that it is the *radiative forcing* from enhanced GHGs that is being offset by solar reduction in G1

Done.

P1 L8: Sentence beginning "In G1,..." - consider starting with "We show that ..." to distinguish your results from prior results

Done. Also added clarification about sign of net forcing.

P1 L19: Consider replacing "compensated for" with "counteracted". Also add a suitable reference at the end of this sentence

Done.

P1 L23: Sentence beginning "since reflecting sunlight would affect ..." is ambiguous. Explain why solar geoengineering leaves residual warming at high latitudes – at least give the primary theories – e.g. more sunlight in the tropics on average – with a suitable reference (e.g. Kravitz et al., 2013)

This sentence has been rewritten, and a reference has been added. The rest of the paragraph has also been revised, to maintain a logical flow.

P2 L12: Replace "subtropical" with "tropical" - Haywood et al identified Sahelian drought as a concern of hemispheric geoengineering and did not look at the subtropics

Done.

P2 L14: Add a suitable reference to the last sentence of this paragraph – which study explicitly identifies an ITCZ shift following a symmetric SAI application?

We now refer to Smyth et al., 2017, which studied ITCZ shifts in G1 (like this study does). A solar constant reduction is an example of a symmetric solar geoengineering deployment. Space mirrors would effectively cause a solar constant reduction.

P2 L15: You say "ITCZ shifts are closely related to the meridional transport of energy by the atmosphere". This is irrefutable, although you should add an appropriate reference, but only tells half the story. Add more discussion about the relative importance of ocean heat transport here, and its ability to control ITCZ position. The following references may be useful: Green and Marshall (2017), Hawcroft et al (2016), Haywood et al (2016), Hwang et al (2017), Marshall et al (2014)

We have added a paragraph discussing the relationships of the ITCZ to the atmosphere and ocean energy transport to the introduction, citing some of these and some other references.

P2 L30: The aim of G1 is not to "keep global mean temperature at approximately preindustrial levels" as you say – be more explicit about the simulation design

We now clarify that the experiment is designed to maintain energy balance at the top of atmosphere.

P4 L5: Define $\nabla.F_L$ at first use (i.e. latent energy transport)

We now define $F_L$, and also clarify $F_L$ and $F_M$ as horizontal fluxes to distinguish from the surface fluxes.

Figs 1 and 2: Consider also changing the y-units to 'poleward energy transport' rather then 'northward energy transport' to assist comparisons. Lastly, I'd suggest putting 'Latitude (N)' as the x-title rather than 'Latitude'

These suggested changes to the figures have been made.

P7 L2: Add a suitable reference for the DSE response to high-latitude warming

We have added reference to Hwang et al. (2011) here, and we also now refer back to Figures 1 and 2.

Eqn 3: This equation is valid for temperature in units in Kelvin, whereas your plots give temperature in units of ºC – pick one for consistency and use throughout the manuscript – I'd personally go with the SI unit K

We now use K consistently throughout the paper.

P8 L14: Sentence "This leaves the differing spatial patterns of forcings as the only possible explanation" should have a caveat that meridional ocean heat transfer is negligible at high-latitudes (e.g. Fig. 6 in Hawcroft et al 2016)

We have now cited this figure as well as the reduction in Atlantic Meridional Overturning Circulation strength found by Hong et al. (2017) and a lack of increased energy input to the tropical ocean as reasons not to expect an increase in oceanic heat transport to the poles.

P8 L30: I recommend that the authors explicitly look at changes to meridional ocean heat transfer in the G1 and abrupt4XCO2 simulations using the methodology of Hawcroft et al (2016). Whilst the caveat about ocean heat transport in P8 L30 is appreciated, explicity looking at changes to ocean heat content / transport would significantly improve the manuscript whilst not altering the primary results

We have expanded on the possible reasons for the poor correlation between the ITCZ shift and cross-equatorial energy flux in abrupt4xCO2 in the revised manuscript. See response to Reviewer 2 major comment 1, which had some suggested reasons for this.

Fig. 4: Consider adding the correlation coefficients to the the respective figures

They are already in this figure, in the titles. We have reduced the number of decimal places from 3 to 2, following Reviewer 2's suggestion.

P9 L2: Reference the Haywood et al (2013) study at the end of "shifts toward the warmed hemisphere"

We decided to delete this sentence, since it lacked clarity. We now reference the Haywood et al. (2013) study at the beginning of the next sentence.

P9 L2: Change "It implies" to "The ITCZ shifts in Fig 4b imply"

Done.

P10 L10: Change "40 ºN" to "40 ºN/S"

Done.

P10 L11: Add a space between piControl and (Figure 5c)

Done.

P10 L12: Yes there are strong correlations, but that doesn't mean the EBM is doing a good job! For instance, for abrupt4XCO2 the EBM predicts a negative poleward energy transport anomaly at 40 ºN for 6 out of the 8 models where the GCMs give a positive anomaly. This issue should be discussed and not glossed over.

The EBM simulations have all been redone (see response to Reviewer 2 major comment 2). Now that humidity is being correctly simulated in the EBM, the EBM does a somewhat better job of reproducing the GCM energy transport changes. For example, the EBM now gives positive energy transport anomalies at 40 ºN/S for all models in Figure 5c. The discussion of Figure 5 has been updated to address the recent changes.

Fig. 5: A minor suggestion - put "40 ºS/N" into the plot titles for b) and c)

Done.

Fig. 6: In the caption and the titles note that it is the *northward* energy transport that you are plotting

Done.

P13 L5: Again – I urge you to explicitly assess oceanic heat uptake in these simulations – you say that the results of the EBM imply that oceanic heat uptake differs between the GCMs – it would not be difficult to assess this hypothesis and would add value to your results

This entire paragraph has been rewritten because of the redone EBM simulations; while the conclusion that clouds are the largest source of inter-model spread has not changed, the spread in the other experiments is somewhat different from before. We now mention here that the spread in this term could represent an atmospheric response to cross-equatorial ocean energy transport. We did not mean that the results of the EBM are the best way to make inferences about oceanic heat uptake—in fact we calculated the net energy flux into the ocean (really into the surface, but over land it is negligible) in order to run this experiment.

Explicitly calculating the ocean energy storage and transport would not be trivial, because it would require obtaining and analyzing output from the ocean component of the models, which

we have not done before. In any case, Hong et al. (2017) have already done a lot of analysis in this direction.

P13 L7: Sentence beginning "The impact of the solar forcing term ..." is very wordy and does not read well – rephrase

This sentence was eliminated in the rewrite of this paragraph.

P13 L25: "has" → "have"

Done.

P13 L27: Sentence beginning "Models with a greater negative change..." - include an example

Added MPI-ESM-LR as an example.

P13 L30: I would also use Fig. 6 in Hawcroft et al (2016) to argue your point that poleward energy transport changes in are likely not due to oceanic heat transport changes – i.e. more background meridional heat transport in the atmosphere than the ocean

We have cited the Hawcroft figure (and the Hong et al. 2017 paper) at the earlier place you suggested it, but we have deleted this sentence. The argument we originally made here was incorrect, because the surfaceFlux experiment is about the response of the atmospheric heat transport to the ocean heat transport and doesn't necessarily tell us anything about what the ocean heat transport itself is doing.

P14 L8: You should add caveats that the EBM does not get the sign of the energy transport right in the Northern Hemisphere (Fig. 5c). I would seriously consider removing Fig. 8A and associated analysis due to this issue as it indicates the EBM is missing the point and any analysis is compromised

Again, the fixed EBM now does get the sign of the energy transport right in the Northern Hemisphere. The discussion of Figure 8A has been revised to account for the redone EBM simulations.

P16 L18: Following "poleward atmospheric energy transport decreases" refer to Figs 1a,d

Done.

**Reviewer 2**

**Overview**

Overall, this manuscript makes an interesting contribution to the physical climate's response to solar radiation management geoengineering. The analysis framework is centered on moist static energy transport, which has been a valuable way of understanding other climate perturbation (e.g., from carbon dioxide and other anthropogenic radiative forcing). Bringing this perspective to the geoengineering simulations is helpful to understand the ways in which the reduced solar constant is not perfectly offsetting the regional climate changes from increased carbon dioxide, even if the global-mean surface temperature is near zero. The conclusions concerning the critical role of the spatial pattern of the forcing (vs. the possibility that radiative feedbacks differ between solar and greenhouse gas forcing) is nice to broadcast clearly, as the abstract does.

I think the manuscript is suitable for publication, but there are some important caveats about the analysis that need to be added and the authors need to alter how they perturb the moist EBM to be more consistent with what we know about the radiative forcing of carbon dioxide (at least the global mean part).

This review, particularly the scrutiny of our methods for imposing the $CO_2$ radiative forcing in the EBM, was extremely helpful to us as it helped us to find a bug in our implementation of the Hwang and Frierson (2010) EBM and to finally find out why we could not originally reproduce one aspect of their results. The bug has been fixed and the EBM experiments have all been re-run and the associated figures replaced. Details of the bug are described below. Thank you also for pointing out interesting papers in the literature that help to better contextualize our analysis and understand its limitations.

**Major Points**

**1) ITCZ shifts: recent literature**

There are a few relevant new publications that the authors should engage with. Seo et al. (2017) showed that the cross-equatorial energy flux does not always account for ITCZ shifts in GCM simulations of carbon dioxide warming. It's fine to continue to use this as a first cut toward understanding the simulations' behavior, but its limitations need to be acknowledged. It may be that the non-zero intercept in Fig. 4a comes about for a reason discussed there: the gross moist stability (energetic stratification) can also change.

We now discuss possible reasons for discrepancies between the cross-equatorial energy flux and ITCZ shifts when discussing Fig. 4a.

Viale and Merlis (2017) analyzed simulations with solar forcing and carbon dioxide forcing and found the ITCZ shifted a different amount. They interpreted this result through the refined energy flux equator argument described by Bischoff and Schneider (2014). The result in Viale

and Merlis (2017) is in line with the typical GCM results here: more sensitivity to carbon dioxide than solar forcing in ITCZ shifts, so a poleward shift in geoengineered climate. This comes about by a different spatial pattern in the radiative forcing (not differences in radiative feedbacks).

We now mention the results of the Viale and Merlis (2017) paper when discussing the multi-model mean ITCZ shift in Section 2.1.

Last, the TRACMIP simulations (Voigt et al., 2016) have examples of radiatively forced warming where the ITCZ shift does not follow the energy flux equator. I've seen this shown in conference talks, though it may not have been published yet.

This is still not published (personal communication from Michela Biasutti). We will add a citation if it is published before the final deadline for changes to our paper.

I think this literature needs to be invoked rather than ocean heat storage (p. 8 L30) for the breakdown in abrupt 4x simulations: the ocean heat storage is already taken into account through the surface fluxes. That cannot explain the why the cross equatorial energy fluxes in the atmosphere are uncorrelated to the ITCZ shift.

We now mention the Seo et al. and Bischoff and Schneider papers when discussing this breakdown, as part of a broader discussion of the limitations of the cross-equatorial energy transport/energy flux equator proxies for the ITCZ position.

**2) Moist EBM analysis**

First, I appreciate that the methodological details are described in a greater detail than in previous papers that use this approach and the authors have made their codes available.

   * Eqn. B1 has a mistake (see North 1975 and subsequent diffusive EBM papers). There is a missing $(1 - x^2)$:

$$\frac{\partial}{\partial x}\left[(1 - x^2)\frac{\partial MSE}{\partial x}\right]$$

I hope this is a typesetting mistake and not a problem with the implementation of the EBM used for the analysis.

Thank you for pointing this out. The factor of $(1-x^2)$ has in fact always been (implicitly) in the EBM code; the error in the manuscript came from us misinterpreting the code from Yen-Ting Hwang that was written in $\theta$ rather than $x$ coordinates. There is also a factor of the square of the radius of the earth that was left out of the equation (absorbed into D in the North papers). In the revised manuscript, for clarity, we have written this equation in both $\theta$ and $x$ coordinates, to help alleviate any confusion about where the $(1-x^2)$ comes from.

* The convention of OLR = $aT_s$ - $b$ is confusing when other EBM literature has the opposite use of $a$ and $b$ (e.g., OLR = $a + bT_s$), but I understand the desire for continuity with Hwang and Frierson (2010).

We have chosen to retain the $aT_s$ - $b$ convention in the revised version.

* I think it's valuable to try to set the record straight concerning how Hwang and Frierson (2010) described the change they made to do the warming EBM simulations. I thought they made a terminology mistake, saying that they re-fit $b$ "to capture the change in climate sensitivity", which would be a change in $a$. Changing just $b$ is reasonable as you can think of that as a change in radiative forcing (e.g., Rose et al., 2014). This manuscript, however, suggests both $a$ and $b$ must be changed to simulate altered climates. I don't follow that, as Rose et al. (2014) and others have done moist EBM simulations that are just changing $b$ and have changes in energy transport. The values of a don't seem to change too much, generally near 2 W/m$^2$/K as expected from the sum of temperature and water vapor feedbacks, so I'm confused.

See response to next comment.

* Again, I want to praise the authors for disclosing the methodological details. I have not seen the parameter values for a and b when using linear fits to GCMs before (Table 4). However, the results are troubling and perhaps earlier work would have attracted scrutiny if this had been disclosed. If we think of $b$ as representing the carbon dioxide radiative forcing (change in OLR with unchanged surface temperature), the fits imply carbon dioxide radiative forcing that varies across models from 0 to 16 W/m$^2$ (just considering the abrupt 4x column vs. the control). This is a problem! It also affects the G1 simulations.

Thank you for bringing up this issue, as it resulted in us finding and fixing a bug in our implementation of the EBM that affected all the EBM simulations. Hwang and Frierson (2010) did in fact re-fit only $b$ instead of $a$ and $b$ (Dargan Frierson confirmed this). The reason we assumed that they didn't was that we could not get much of a response of energy transport to the "greenhouse" experiment when changing only $b$. It turned out that when re-implementing the EBM in Python, we neglected to convert saturation vapor pressure from hPa to Pa, resulting in specific humidity being too low by a factor of 100. So, we were effectively running a dry EBM instead of a moist EBM. Now that this problem has been fixed, we get changes in MSE transport when changing only $b$ that are of similar magnitude to those in the Hwang $et$ $al.$ papers. Before the moisture bug was fixed, changing only $b$ amounts to a uniform reduction in OLR everywhere, which has no effect on the energy transport (so this methodology won't work in a dry EBM). But when humidity can adjust to temperature, it goes up much more in the tropics than the poles responding to a uniform initial OLR perturbation, leading to an increase in meridional energy transport as a result of the stronger greenhouse effect.

We re-ran all of the EBM simulations with the corrected moisture issue, also having fixed a more minor bug where it had been assumed the GCM latitudes are evenly spaced (in fact most GCMs

use non-even Gaussian grids), and re-fitting only $b$ instead of $a$ and $b$ in the all-perturbation and greenhouse experiments. The results are qualitatively similar in most respects to those in the previous version of the manuscript, and our overall conclusions have not changed. The agreement between the EBM and GCMs in Figure 5 has improved somewhat. The plot axes in Figures 5 through 8 had to be expanded because the EBM moves more energy now, and the text has been changed to reflect the changes in methodology and results where necessary.

Regarding the value of the $CO_2$ forcing, the values of $b'$ where $a$ has been fixed when re-fitting are now included in Table 4 (and the old values of $a'$ and $b'$ deleted). Comparing $b'$ to $b$ in this table shows that the radiative forcing values are now all around 7-9 W/m$^2$ for G1; for abrupt4xCO2 they are a little higher in some models (as much as 13 W/m$^2$ for CESM) but this doesn't seem entirely unreasonable because clear-sky LW feedbacks such as water vapor and lapse rate feedbacks are included. (Also, $b'$ is greater than $b$, as it should be with the OLR $= aT_s - b$ convention; when re-fitting $a$ and $b$, not only were the values of radiative forcing wildly divergent, as you pointed out, but they were all *negative*.)

       * The last important thing that should be noted is that the approach of prescribing TOA perturbations (e.g., from cloud radiative effect changes) in the moist EBM does not capture the fact that feedbacks are interacting with each other (Feldl et al., 2017) and with the changing energy transport (Rose et al., 2014; Merlis, 2014; Rose and Rayborn, 2016). There's still value in this way of diagnosing things (partially interactive), but it's not the full story and may help explain why EBM is changes less than the GCM.

       We now note these limitations, as well as the fact that the EBM does not capture the spatial structure of $CO_2$ forcing, in the 2$^{nd}$ paragraph of Section 3. Note that the poleward energy transport change in the EBM is now greater than that for the GCMs in G1 and for the northern hemisphere in abrupt4xCO2 in the redone EBM experiments (see Figures 5b,c).

**3) Forcing estimates**

The authors describe in conclusions that the radiative forcing for carbon dioxide is not uniform, but this seems important enough to come earlier when describing the EBM set up (p. 10 L1).

       * The authors cite only Zelinka and Hartmann (2012) (p. 15) when discussing the extratropical response to carbon dioxide. Those authors neglected the structure of carbon dioxide, which was subsequently included in Huang and Zhang (2014). They also looked at the same CMIP5 abrupt 4x simulations, so it's important to confirm consistency.

Thank you for pointing out the role of the non-uniform $CO_2$ forcing in the response of energy transport to $CO_2$ increases. The non-uniform forcing not captured by the EBM when re-fitting only $b$. We now cite the Huang and Zhang paper and mention the non-uniform forcing in the discussion of the "greenhouse" EBM experiment in Section 3.3.1.

\* Other authors (e.g., Feldl and Bordoni, 2016) have taken the approach of diagnosing the forcing, including its spatial structure, from fixed SST simulations ("troposphere adjusted forcing"). Given my concern about the linear fit estimate of $b$ values above, I think this is a superior way to drive the moist EBM. Another alternative would be to take previously published global-mean adjusted forcing estimates and use those—at least all the carbon dioxide forcing would be near 7 W/m$^2$.

Having re-run all the EBM simulations with corrected moisture transport and re-fitting only $b$ and not $a$, we think we have adequately addressed the concerns about accuracy of the $CO_2$ forcing. It is true that this does not capture the spatial structure of the $CO_2$ forcing, but there are no obvious, workable alternatives. Using forcing estimates from fixed-SST simulations would include rapid adjustments from clouds and other variables in the "greenhouse" experiment. This experiment was intended to simulate the effects of the enhanced greenhouse effect, including water vapor, lapse rate and Planck feedbacks, but not cloud feedbacks. If cloud feedbacks were included in the greenhouse experiment, they would be double-counted when the SW and LW cloud effects were also included (as in the "all_G1" and "all_4xCO2" experiments), and it would be hard to distinguish between the effects of $CO_2$ forcing and changes in clouds. We also considered re-fitting $b$ in individual latitude bands instead of globally, but since we can only use clear-sky OLR for the fit, this would miss the portion of the spatial structure of radiative forcing that is caused by pre-existing clouds (thanks to Dargan Frierson for pointing this out).

**Minor Points**

p. 3 L19: "moisture fluxes" terminology is not ideal for the latent surface fluxes, given that meridional energy and moisture fluxes are discussed later.

"Moisture" here has been changed to "latent heat".

Fig. 3: how close is the global-mean surface temperature change to zero in these simulations? I understand this is discussed in other papers using G1, but it would be helpful to at least state a representative number (e.g., within 0.2 K).

Global mean temperature changes have been added to Table 1.

Fig. 3 caption: presumably these are zonal \*and annual\* mean \*surface air\* temperature changes? I didn't see this in the main text either. Likewise for saturation vapor pressure.

It has now been clarified that these are surface air temperature and surface specific humidity. A clarification that all months are included has been added to the description of the averaging procedure in the first paragraph of Section 2.

p. 8 L27, and elsewhere: a big excessive on the number of digits for $r$.

The number of digits for $r$ has been reduced from 3 to 2 throughout the paper.

Fig. 5: plot 1-to-1 lines here

Done. Also, the plot aspect ratio has been changed to 1:1.

p. 16 L6: the ocean heat uptake is weighted toward subpolar oceans (vs. a uniform uptake which wouldn't affect extratropical atmospheric energy transport)

We now make this clarification in the text.

**Specific Comment 1**

P4 L10-13: The "unphysical result" problem stated in your manuscript is due to a postprocessing (CMORizing) error in the first G1 realization (r1i1p1) from BNU-ESM, there is no such problem in the second G1 realization (r2i1p1). You could download all BNUESM CMIP5/GeoMIP results at http://esg.bnu.edu.cn/thredds/esgcet/catalog.html. If you have problems please let us know.

The r2i1p1 realization of G1 in BNU-ESM still has the problem that the global mean temperature is not adequately maintained at preindustrial levels, which is important to our analysis. After this comment was posted, the BNU modeling group provided us with the output from a new G1 realization, r3i1p1, which did a good job of maintaining global mean temperature. However, there was not enough time to incorporate this model into our analysis for this revision. We have removed the reference to the "unphysical result" in the text.

[revised manuscript text omitted]